# Toward Efficient Multi-Agent Exploration With Trajectory Entropy Maximization

**Tianxu Li**[1,2]  **Kun Zhu**[1,2,*]
[1]College of Computer Science and Technology, Nanjing University of Aeronautics and Astronautics, China
[2]Collaborative Innovation Center of Novel Software Technology and Industrialization
{tianxuli, zhukun}@nuaa.edu.cn

## Abstract

Recent works have increasingly focused on learning decentralized policies for agents as a solution to the scalability challenges in Multi-Agent Reinforcement Learning (MARL), where agents typically share the parameters of a policy network to make action decisions. However, this parameter sharing can impede efficient exploration, as it may lead to similar behaviors among agents. Different from previous mutual information-based methods that promote multi-agent diversity, we introduce a novel multi-agent exploration method called Trajectory Entropy Exploration (TEE). Our method employs a particle-based entropy estimator to maximize the entropy of different agents' trajectories in a contrastive trajectory representation space, encouraging visitations of diverse trajectories and efficient exploration. This entropy estimator avoids challenging density modeling and scales effectively in high-dimensional multi-agent settings. We integrate our method with MARL algorithms by deploying an intrinsic reward for each agent to encourage entropy maximization. To validate the effectiveness of our method, we test our method in challenging multi-agent tasks from several MARL benchmarks. The results demonstrate that our method consistently outperforms existing state-of-the-art methods.

## 1 Introduction

Multi-Agent Reinforcement Learning (MARL) has gained considerable attention in recent years for its potential to solve multi-agent tasks, such as multiplayer video games (Vinyals et al., 2019) and traffic light control (Wu et al., 2020). MARL facilitates effective cooperation by training multiple agents together to maximize overall team performance. However, developing efficient cooperative policies for challenging multi-agent tasks remains difficult due to the constraints of partial observation and the need for scalability. A widely adopted solution to these challenges is the Centralized Training with Decentralized Execution (CTDE) framework (Lowe et al., 2017), in which each agent takes actions based on local observations while being trained with access to global information to ensure robust and stable performance.

In the CTDE framework, each agent learns its own decentralized policy. Training numerous policy networks can be inefficient. To address this, the parameter sharing technique is typically employed, allowing all agents to use the same policy network parameters to make action decisions. This approach significantly reduces the number of required parameters, thereby lowering computational complexity and speeding up the training process. Additionally, parameter sharing facilitates the experience sharing among agents during centralized training, which not only helps in learning a robust and stable policy but also increases overall learning efficiency (Wang et al., 2020b).

Taking advantage of these benefits, many MARL algorithms have integrated the parameter sharing technique, including value-based methods (Iqbal et al., 2021; Yang et al., 2021; Wang et al., 2020a; Sunehag et al., 2018; Rashid et al., 2018) and policy gradients (Ma et al., 2021; Wang et al., 2020d; Ndousse et al., 2021; Zhang et al., 2021; Yu et al., 2022; Kuba et al., 2021). However, when agents share the same policy network parameters, they often learn homogeneous behaviors, as they tend

---

*Corresponding author.

to exhibit similar behaviors when faced with similar observations (Hu et al., 2022). This limits the emergence of multi-agent diversity and hinders efficient exploration. Complex multi-agent tasks typically require extensive exploration and varied policies among agents. For example, in a football game, where agents must work together to score, uniform policies might lead to competition for the ball, resulting in ineffective play. To succeed, agents need to learn diverse strategies and assume different roles to effectively pass the ball and score.

To solve the problem, previous works (Jiang and Lu, 2021; Li et al., 2021; Charakorn et al., 2023; Jo et al., 2024) typically promote multi-agent diversity based on identity-aware trajectories in a fully-supervised manner. They maximize the mutual information between trajectories and agent identities to differentiate the trajectories of different agents based on their identities. However, despite their success, these methods are prone to overfitting and falling into local optima, as agents tend to revisit familiar trajectories rich in identity information, hindering efficient exploration. Consequently, the agents' trajectories may become overly focused on aligning with their identities.

In this paper, we introduce a novel exploration method called Trajectory Entropy Exploration (TEE) designed to enhance multi-agent diversity while ensuring efficient exploration. Unlike previous methods, our method does not rely on mutual information or trajectory discriminators. The intuition is that agents must thoroughly explore the environment to visit states where they might obtain rewards. To achieve this, our method focuses on maximizing the entropy of different agents' trajectories. Since directly maximizing entropy in the high-dimensional trajectory space is intractable due to the unknown trajectory density model, we employ a nonparametric particle-based entropy estimator (Singh et al., 2003; Beirlant et al., 1997). This estimator is asymptotically unbiased for entropy by calculating the mean Euclidean distance between a particle and its neighbors. In multi-agent settings, to make these distances meaningful, we build a trajectory representation space through encoding the trajectory space into a low-dimensional representation using contrastive learning (Chen et al., 2020). Contrastive learning is a commonly used method to learn useful representations, encoding similarities and dissimilarities among sample instances (Chen et al., 2020).

The contributions of this work are summarized as follows: First, we efficiently distinguish trajectory representations of different agents using contrastive learning. Since agents adopting parameter sharing may induce similar trajectory samples, it is intractable to directly use the vanilla contrastive learning. We further introduce an identity representation for each agent to distinguish trajectory representations of different agents. The identity representations serve as medium variables to contrast different trajectory samples. Unlike fixed agent identities used in previous works, identity representations are vectors, consisting of learnable parameters, to linearly classify trajectory representations for minimal contrastive learning loss. Second, to encourage multi-agent diversity, we maximize the trajectory entropy using a nonparametric particle-based entropy estimator in the learned trajectory representation space. We adapt the entropy estimator to multi-agent settings by calculating the average distance between a particle and its $k$ nearest neighbors, achieving more stable and robust empirical results. Third, we integrate our method into MARL algorithms by deploying an intrinsic reward, based on the entropy estimator, encouraging agents to maximize entropy. Fourth, we demonstrate the effectiveness of our method through experiments on various challenging multi-agent tasks, where TEE significantly outperforms existing state-of-the-art MARL algorithms.

## 2 BACKGROUNDS

We model fully cooperative multi-agent tasks as a Decentralized Partially Observable Markov Decision Process (Dec-POMDP) (Oliehoek and Amato, 2015), defined by the tuple $\langle A, S, U, P, R, O, \Omega, \gamma \rangle$. Here, $A = \{1, \ldots, |A|\}$ represents a set of $|A|$ agents, $s \in S$ denotes the environment state, and $U$ is the set of possible actions. At every time step, agent $a$ receives an observation $o^a \in \Omega$ based on the observation function $O(s, a)$ and choose an action $u^a \in U$. The actions of all agents combine to form a joint action $\boldsymbol{u}$. The environment then transitions to a new state $s'$ according to the transition function $P(s' \mid s, \boldsymbol{u})$. At the same time, agents receive a shared reward $r = R(s, \boldsymbol{u})$ from the environment. The reward discount factor is represented by $\gamma \in [0, 1)$. Each agent's trajectory, composed of its observation-action history, is denoted as $\tau^a \in \mathcal{T}$. Each agent learns a decentralized policy $\pi^a (u^a \mid \tau^a)$, which together form a joint policy $\boldsymbol{\pi}$. The goal is to maximize the joint action-value function $Q^{\boldsymbol{\pi}}(s, \boldsymbol{u}) = \mathbb{E}_{s_{0:\infty}, \boldsymbol{u}_{0:\infty}} \left[ \sum_{t=0}^{\infty} \gamma^t r_t \mid s_0 = s, \boldsymbol{u}_0 = \boldsymbol{u}, \boldsymbol{\pi} \right]$.

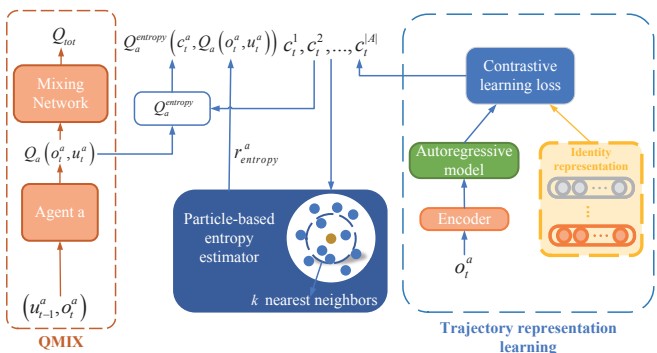

Figure 1: Diagram of our proposed TEE. Our method trains the encoder and the autoregressive model by minimizing the contrastive loss to learn an abstract trajectory representation space from trajectories of different agents. We deploy an intrinsic reward function based on a particle-based entropy estimator in the MARL algorithm such that we can train the policies of agents towards maximizing the trajectory entropy, encouraging multi-agent diversity.

## 3 TRAJECTORY ENTROPY EXPLORATION

One of the common approaches to encourage multi-agent diversity is to maximize the mutual information between trajectories and agent identities (Jiang and Lu, 2021; Li et al., 2021). However, these works share a limitation that the agents are likely to prefer known trajectories, containing more identity information, than novel trajectories, resulting in inefficient exploration. Consider a variational reward $r(\tau, i) = \log q_\theta(i \mid \tau) - \log p(i)$ that maximizes the mutual information objective. $q_\theta(i \mid \tau)$ is a variational distribution, parameterized by $\theta$, which approximates the true posterior probability of the agent identity $i$ conditioned on the trajectory $\tau$. $p(i)$ is assumed to be a uniform distribution, so $-\log p(i) = \log |A|$ is a constant, where $|A|$ is the number of agents. When agents visit known trajectories that have been successfully discriminated by $q_\theta(i \mid \tau)$, i.e., $q_\theta(i \mid \tau) \rightarrow 1$, the variational reward is $r_{\text{known}} = \log 1 + \log |A| = \log |A|$. However, when agents visit new trajectories, the variational reward is $r_{\text{new}} = \lim_{q_\theta(i|\tau) \rightarrow 0} \log q_\theta(i \mid \tau) + \log |A| = -\infty$. We note that when agents visit known trajectories, they can achieve more rewards. We provide a more detailed theoretical analysis of this limitation in Appendix B. To resolve this limitation, in this paper, our aim is to encourage multi-agent diversity by maximizing the entropy of trajectories of different agents in an abstract representation space, unlike the previous mutual information maximization method. First, we map the trajectory space to a latent contrastive representation space with a contrastive learning method. Then, we propose a novel nonparametric method to maximize the trajectory entropy by introducing per-agent intrinsic rewards.

### 3.1 LEARNING CONTRASTIVE TRAJECTORY REPRESENTATIONS

We first construct a trajectory representation space to enable the proper functioning of the nonparametric particle-based entropy estimator. To achieve this goal, we resort to contrastive learning, which has shown great promise in learning meaningful representations in RL, as demonstrated in recent works (Laskin et al., 2020; Stooke et al., 2021). However, the vanilla contrastive learning may not be immediately applicable in our multi-agent setting, as the policy network parameter sharing may lead to similar trajectories. By contrasting these similar trajectory samples against each other, the vanilla contrastive learning may fail to distinguish trajectory representations of different agents based on distance between them, which is necessary for entropy maximization to work. To solve this issue, we introduce a learnable identity representation for each agent and instead contrast the trajectory samples with the identity representations to pull the trajectory samples with the same identity representations together while pushing apart those with different identity representations in the trajectory representation space.

Concretely, we encode trajectories into the trajectory representations $c_t^a$ by encoding observations into latent embeddings $z_t^a = g_{\theta_e}(o_t^a)$ with a non-linear encoder $g_{\theta_e}$. These embeddings are then summarized with an autoregressive model $g_{\theta_g}$, i.e., $c_t^a = g_{\theta_g}(z_{\leq t}^a)$, which alleviates the non-stationary

issue caused by partial observations. We further denote the trajectory encoder as $g_\theta = \{g_{\theta_e}, g_{\theta_g}\}$. Let the set $\mathcal{C} = \left\{ c_t^{a'} \right\}_{a'=1}^{|A|}$ represents the trajectory representations of all agents at time step $t$, and let $d^a \in \mathbb{R}^{\mathbb{H}}$ denote the identity representation for agent $a$. To train the trajectory encoder $g_\theta$ in order to learn distinguishable trajectory representations, we minimize a contrastive learning loss, or an InfoNCE loss (Oord et al., 2018),

$$\mathcal{L}_N = - \mathop{\mathbb{E}}_{(d^a, \mathcal{C}) \sim \mathcal{D}} \left[ \log \frac{f\left(c_t^a, d^a\right)}{\sum_{c_t^{a'} \in \mathcal{C}} f\left(c_t^{a'}, d^a\right)} \right] \tag{1}$$

where $f(c_t, d) = \exp\left(c_t^T d\right) \in \mathbb{R}$. We use the dot product similarity $c_t^T d$ to measure the distance between the identity representation and the trajectory representation. The goal of the contrastive learning loss is to guarantee that the identity representation $d^a$ remains close with its corresponding trajectory representations $c_t^a$ while being far away from the trajectory representations of other agents in $\mathcal{C} \setminus \{c_t^a\}$. As a result, the identity representations of all agents are uniformly distributed on the trajectory representation hypersphere with their corresponding trajectory representations scattered around them, leading to distinguishability among trajectory representations. Note that the identity representations introduced in our method linearly classify the trajectory representations of different agents for the minimal contrastive learning loss.

## 3.2 NONPARAMETRIC ENTROPY MAXIMIZATION

Prior work (Hazan et al., 2019) typically uses density estimation to maximize entropy. However, in high-dimensional multi-agent settings, it is intractable and non-trivial to estimate the density. In practice, a nonparametric particle-based entropy estimator (Singh et al., 2003; Beirlant et al., 1997), which has been extensively studied in statistics (Jiao et al., 2018), is employed in our method to achieve entropy maximization of trajectory representations across different agents. This estimator quantifies the sparsity of data distribution by measuring the distance between a data point and its $k$-th nearest neighbor.

We next describe the details of the particle-based entropy estimator within our method. In multi-agent settings, each trajectory representation learned by $g_\theta$ is considered as a particle. Specifically, for a collection of trajectory representations $\{c_t^a\}_{a=1}^{|A|}$ induced by all agents, the particle-based entropy estimator is formulated as:

$$\mathcal{H}(c_t) = -\frac{1}{|A|} \sum_{a=1}^{|A|} \log \frac{k}{|A| \mathrm{v}_a^k} + b(k) \propto \sum_{a=1}^{|A|} \log \mathrm{v}_a^k, \tag{2}$$

where $b(k)$ acts as a bias correction based on the hyperparameter $k$, and $\mathrm{v}_a^k$ represents the volume of a hypersphere with a radius equal to $\left\| c_t^a - (c_t^a)^{(k)} \right\|$,

$$\mathrm{v}_a^k = \frac{\left\| c_t^a - (c_t^a)^{(k)} \right\|^{|A|} \cdot \pi^{|A|/2}}{\Gamma\left(|A|/2 + 1\right)} \tag{3}$$

where $(c_t^a)^{(k)}$ denotes the $k$-th nearest neighbor of $c_t^a$ in the set $\{c_t^a\}_{a=1}^{|A|}$, $\|\cdot\|$ refers to the Euclidean distance, and $\Gamma$ is the gamma function. Essentially, $\mathrm{v}_a^k$ indicates the sparsity around each agent's trajectory representation. The entropy estimator $\mathcal{H}(c_t)$ thus measures the average of the sparsity surrounding the trajectory representations of all agents.

Based on the definition of $\mathrm{v}_a^k$, the particle-based entropy estimator in Equation 2 can be rewritten as

$$\mathcal{H}(c_t) \propto \sum_{a=1}^{|A|} \log \left\| c_t^a - (c_t^a)^{(k)} \right\|^{|A|} \tag{4}$$

where the entropy $\mathcal{H}(c_t)$ is proportional to the sum of the logarithms of the distances between each trajectory representation and its $k$-th nearest neighbor. However, in multi-agent settings, we empirically found that the entropy estimator in Equation 4 based on the $k$-th nearest neighbor typically result in learning unstable policies. To address this, we introduce a novel entropy estimator

for multi-agent settings, which averages the distances over all $k$ nearest neighbors for each trajectory representation:

$$\mathcal{H}(c_t) := \sum_{a=1}^{|A|} \log \left( d + \frac{1}{k} \sum_{(c_t^a)^{(j)} \in N_k(c_t^a)} \left\| c_t^a - (c_t^a)^{(j)} \right\|^{|A|} \right), \tag{5}$$

In this formula, $N_k\left(c_t^a\right)$ denotes the set of $k$ nearest neighbors surrounding a trajectory representation $c_t^a$. We also incorporate a constant $d$, set to 1 in all experiments, to enhance numerical stability.

To promote diversity among multiple agents by maximizing the entropy $\mathcal{H}(c_t)$, we can use the entropy as an intrinsic reward $r_{entropy}^a$, where the representation of $o_{t+1}^a$ is considered as a particle contributing to the entropy. Specifically, given a transition $(o_t^a, u_t^a, o_{t+1}^a)$ for agent $a$, we define the intrinsic reward function for agent $a$ as follows:

$$r_{entropy}^a = \log \left( d + \frac{1}{k} \sum_{g_\theta(o_{t+1}^a)^{(j)} \in N_k\left(g_\theta(o_{t+1}^a)\right)} \left\| g_\theta(o_{t+1}^a) - g_\theta(o_{t+1}^a)^{(j)} \right\|^{|A|} \right). \tag{6}$$

Intuitively, this intrinsic reward encourages agents to explore diverse trajectories with larger distances in the trajectory representation space. For the PyTorch-style pseudocode of TEE, please refer to Appendix E. The source code of our method can be found in the supplemental material.

**Differences to mutual information-based methods** It is important to note that our objective differs significantly from previous methods (Jiang and Lu, 2021; Li et al., 2021; Charakorn et al., 2023; Jo et al., 2024), which leverage the maximization of mutual information between trajectories $\tau$ and agent identities $i$ by introducing an intrinsic reward:

$$r_{MI}(\tau, i) = \log q_\theta(i \mid \tau) - \log p(i) \tag{7}$$

where $q_\theta(i \mid \tau)$ represents a variational distribution trained to maximize the likelihood of $(i, \tau)$-tuples stored in the replay buffer. $p(i)$ is a fixed uniform distribution. The mutual information-based intrinsic reward $r_{MI}$ motivates agents to visit trajectories that carry more identity-specific information. In contrast, our trajectory entropy-based intrinsic reward $r_{entropy}^a$ in Equation 6, incentivizes agents to explore a variety of trajectories with greater distances in the trajectory representation space, thereby resulting in entropy maximization.

## 3.3 LEARNING ALGORITHM

In this section, we present how to integrate our algorithm with existing MARL methods. As our method introduces a trajectory entropy-based intrinsic reward for each agent, the agent needs to learn its own decentralized policy independently towards maximizing the intrinsic rewards. We first show how to integrate our method with QMIX (Rashid et al., 2018), a value-decomposition-based MARL method. QMIX co-trains the policies of all agents by optimizing an approximation, $Q_{tot}$, for the joint action-value function $Q^\pi$. QMIX uses a mixing network to monotonically combine the utility functions of all agents (from which the agents' policies are derived) to calculate $Q_{tot}$. Since the policies of all agents are co-trained by QMIX to maximize the shared team rewards, we cannot simply add the individual intrinsic rewards to the shared team rewards to maximize the individual intrinsic rewards for each agent. To solve this limitation, we additionally learn a shared intrinsic utility network, $Q_a^{entropy}$, for each agent. The intrinsic utility network $Q_a^{entropy}$ uses the agent's utility $Q_a(o_t^a, u_t^a)$ as well as the current trajectory representation $c_t^a$ as inputs. To train the intrinsic utility network $Q_a^{entropy}$, we minimize the TD loss using the intrinsic rewards:

$$\mathcal{L}_{TD}^{entropy} = \mathbb{E}_{(o_t^a, u_t^a, o_{t+1}^a) \sim \mathcal{D}} \left[ \left( Q_a^{entropy}\left(c_t^a, Q_a(o_t^a, u_t^a)\right) - y \right)^2 \right],$$
$$where \quad y = r_{entropy}^a + \gamma \bar{Q}_a^{entropy}\left(c_{t+1}^a, \bar{Q}_a\left(o_{t+1}^a, u_{t+1}^a\right)\right). \tag{8}$$

where $\bar{Q}_a^{entropy}$ and $\bar{Q}_a$ are target networks used to stabilize training. For each training iteration, we randomly sample a mini-batch of trajectory data from the replay buffer $\mathcal{D}$. As the intrinsic utility

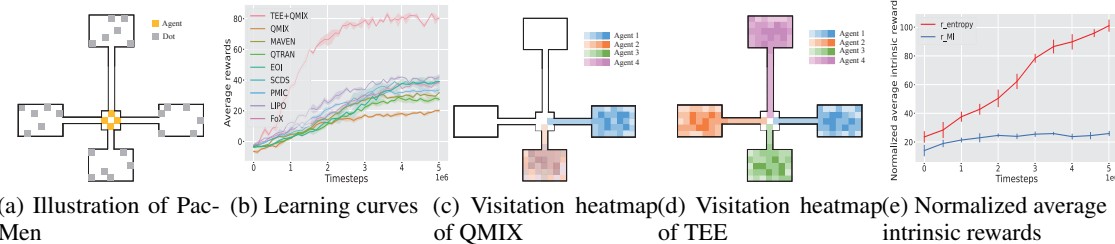

(a) Illustration of Pac-Men    (b) Learning curves    (c) Visitation heatmap of QMIX    (d) Visitation heatmap of TEE    (e) Normalized average intrinsic rewards

Figure 2: Performance comparison between our proposed TEE and baselines in Pac-Men.

network takes the agent's utility $Q_a(o_t^a, u_t^a)$ as an input, the loss function $\mathcal{L}_{TD}^{entropy}$ provides an auxiliary gradient to help train the agent's utility network, leading to trajectory entropy maximization. Since the agent utility in QMIX has no inherent meaning or constraints, our method can be safely integrated with QMIX. Thus, the overall loss function for learning optimal agent policies is:

$$\mathcal{L}_{total} = \mathcal{L}_{TD}^{QMIX} + \beta \mathcal{L}_{TD}^{entropy}, \tag{9}$$

where $\mathcal{L}_{TD}^{QMIX}$ is the TD loss function in QMIX that is used to learn the optimal $Q_{tot}$ and to update the parameters of the agent utility networks with the goal of maximizing team returns. The coefficient $\beta$ adjusts the weight of $\mathcal{L}_{TD}^{entropy}$ in comparison to $\mathcal{L}_{TD}^{QMIX}$.

During training, we alternately train the trajectory encoder and policies of agents. We first sample trajectories from the replay buffer to train the encoder to learn distinguishable trajectory representations by minimizing the contrastive learning loss. Then we calculate the particle-based trajectory entropy estimator based on the learned trajectory representations for policy learning. The policies of agents are trained end-to-end in a centralized manner by minimizing $\mathcal{L}_{total}$, allowing each agent to learn a policy that maximizes both the team returns and the trajectory entropy across different agents. Therefore, our method fosters diversity among agents, effectively solving the limitation of parameter-sharing for efficient exploration. Additionally, our method can be integrated with policy gradient methods. For details on implementing our method with policy gradient methods, please refer to Appendix D.

## 4 EXPERIMENTS

In this section, we examine the performance of our proposed TEE method using challenging multi-agent tasks from Pac-Men, SMAC, and SMACv2 benchmarks, demonstrating its superior effectiveness. We compare TEE against state-of-the-art methods, including value-decomposition methods like QMIX (Rashid et al., 2018) and QTRAN (Son et al., 2019), as well as mutual information-based exploration strategies such as MAVEN (Mahajan et al., 2019), EOI (Jiang and Lu, 2021), SCDS (a variant of CDS (Li et al., 2021) with shared policy network parameters), PMIC (Li et al., 2022), LIPO (Charakorn et al., 2023), and FoX (Jo et al., 2024). Without loss of generality, we present both the mean and standard deviation of performance for our method and the baseline methods, tested with five random seeds. To ensure fairness, consistent hyperparameters and policy network structures are applied across different methods for each multi-agent task, with detailed experimental settings provided in Appendix H.

### 4.1 PAC-MEN

To highlight the effectiveness of our method in fostering multi-agent diversity, we adopt a grid world environment called Pac-Men, depicted in Figure 2a, to evaluate our method against baseline methods. In this environment, four agents start in the central room of a maze, each with limited visibility. Dots are randomly placed in the edge rooms, and the agents must navigate along various paths to collect them. To increase the difficulty, each path to the edge rooms has a different length. Importantly, only the downward path falls within the agents' observable area, making efficient exploration essential.

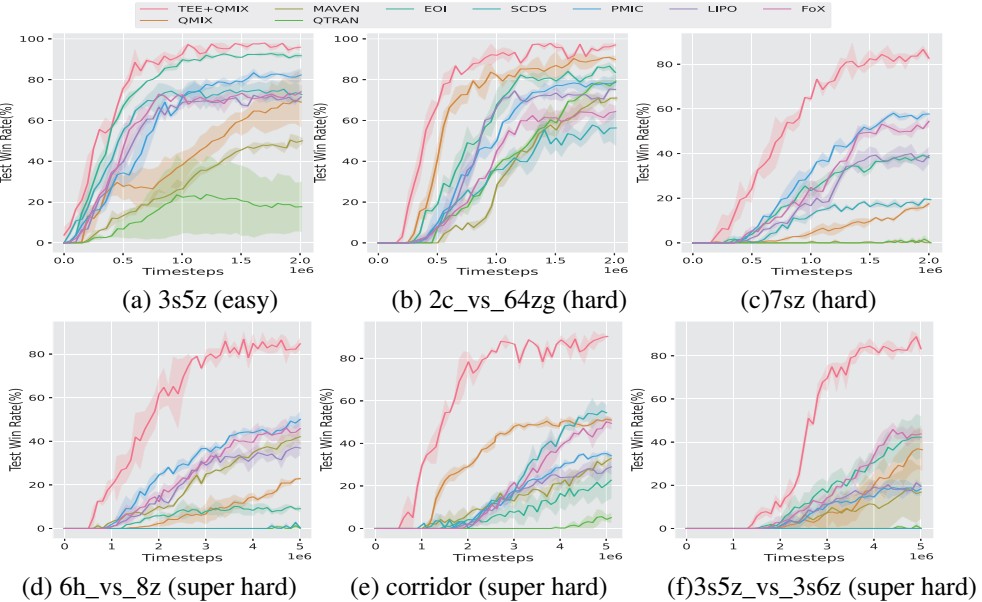

Figure 3: Performance comparison between our proposed TEE and baselines in the SMAC scenarios.

As depicted in Figure 2b, our method demonstrates significant superiority over QMIX and substantially outperforms the other baselines. QMIX fails to learn optimal policies in Pac-Men. For the agents to maximize team returns, they need to disperse to the four edge rooms to collect dots. However, the visitation heatmap for QMIX, shown in Figure 2c, reveals that some agents adopt similar behaviors, converging in the same bottom room. This overlap leads to competition for the same dots, resulting in inefficient cooperation. In contrast, our method, as illustrated in Figure 2d, enables the agents to efficiently learn diverse policies, allowing them to respectively move to different edge rooms. This outcome indicates that entropy maximization effectively encourages the development of diverse strategies. Baseline methods like EOI and SCDS, which aim to maximize the mutual information between trajectories and agent identities, perform similarly but fall short of achieving satisfactory results. We believe this is due to their inadequate exploration, as agents may fail to discover the upward room with the longest path. Figure 2e further compares the intrinsic rewards generated by mutual information-based methods and our entropy maximization method. The results indicate that mutual information-based rewards offer limited incentives, while our entropy maximization-based rewards consistently motivate the agents to explore and learn optimal cooperative policies.

## 4.2 SMAC

After assessing our method in a basic grid world environment, we move on to a more challenging multi-agent benchmark, namely StarCraft Multi-Agent Challenge (SMAC) (Samvelyan et al., 2019). To demonstrate the effectiveness of our method, we examine our method across six SMAC scenarios with increasing difficulty levels: 3s5z (easy), 2c_vs_64zg (hard), 7sz (hard), 6h_vs_8z (super hard), corridor (super hard), and 3s5z_vs_3s6z (super hard). It is important to note that performance comparisons are not valid across different SMAC versions. For our experiments, we use SMAC version SC2.4.10.

The comparisons of performance between our method and the baseline methods in the SMAC scenarios are presented in Figure 3. In the super hard scenarios (6h_vs_8z, corridor, and 3s5z_vs_3s6z), where the enemies are significantly stronger than the agents, our method considerably outperforms the baselines. This suggests that TEE is more effective at exploring cooperative policies by maximizing trajectory entropy. Challenging scenarios impose a high demand on policy diversity to distribute enemies' attacks. Our method successfully learns diverse policies. To demonstrate this, we provide visualization examples of the diverse policies learned by our method in Appendix M. While QMIX performs well in the 3s5z and 2c_vs_64zg scenarios, it struggles to learn effective policies in the more challenging scenarios that demand complex cooperation strategies, whereas our method excels.

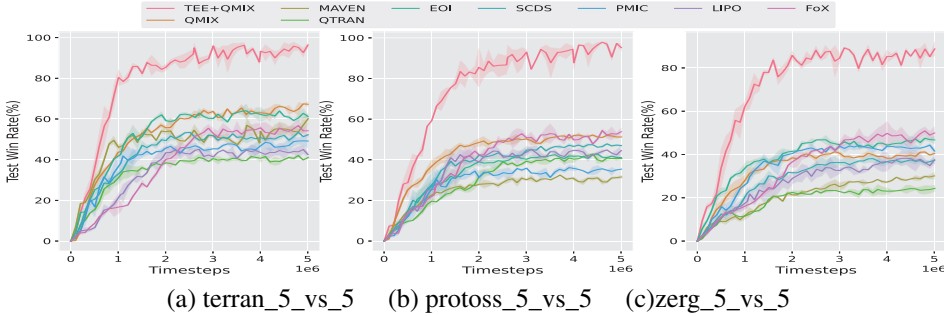

Figure 4: Performance comparison between TEE and baselines in the SMACv2 scenarios.

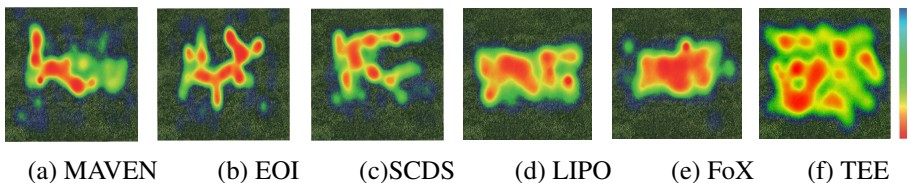

(a) MAVEN     (b) EOI     (c)SCDS     (d) LIPO     (e) FoX     (f) TEE

Figure 5: Visitation heatmaps of different algorithms in the terran_5_vs_5 scenario.

MAVEN proves less effective at exploring cooperative policies, highlighting that the trajectory entropy maximization objective leads to more efficient exploration compared to the strategy of encouraging diverse joint behaviors by MAVEN. While EOI and SCDS deliver promising results in the 3s5z and 2c_vs_64zg scenarios, they fall short in more challenging scenarios. We believe this is due to the strong mutual dependence between trajectories and agent identities, which hinders the exploration of complex cooperative strategies. Similarly, the formation diversity based on mutual information in FoX also encounters this issue.

**Homogeneous behaviors** Our method also shows superior performance in the easy 3s5z scenario, where agents sometimes need to act similarly to master the 'focus fire' trick. This indicates that our method efficiently trades off exploration and exploitation, and does not hinder the learning of homogeneous behaviors that are beneficial for maximizing environmental rewards. For additional evaluations of our method in scenarios that require homogeneous behaviors, please refer to Appendix J.

**Stochasticity and Exploration** Despite the challenging settings in the SMAC scenarios, a notable limitation is the insufficient stochasticity in combat scenarios due to the fixed initial positions of units and team compositions. This limits the ability to fully test the exploration capabilities of MARL algorithms. To overcome this, we adopt the more demanding SMACv2 benchmark (Ellis et al., 2022), which introduces stochastic elements by randomizing start positions and team compositions in each episode.

We examine our method in three SMACv2 scenarios including terran_5_vs_5, protoss_5_vs_5, and zerg_5_vs_5. The experimental results, shown in Figure 4, indicate that our method significantly outperforms the baselines across all scenarios. Notably, QMIX struggles to learn optimal cooperative policies and lacks the necessary exploration to adapt to the stochasticity present in the SMACv2 scenarios. However, when combined with our method, QMIX shows marked improvement in performance, learning more exploratory and diverse policies. Similarly, mutual information-based baselines like MAVEN, EOI, and SCDS tend to get stuck in local optima. We believe this occurs because the mutual dependence between agent identities and trajectories in these methods restricts agents to known trajectories rather than encouraging the discovery of new ones. In contrast, our method consistently explores new trajectories and seeks out exploratory policies. Additionally, we provide agent's visitation heatmaps in Figure 5. The results clearly show that agents trained with the baselines confine their movements to only partial areas, whereas our method motivates the agents to explore all possible states that offer environmental rewards, ensuring thorough exploration of the entire environment.

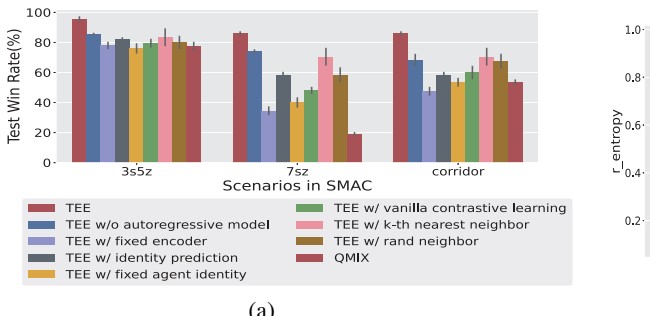 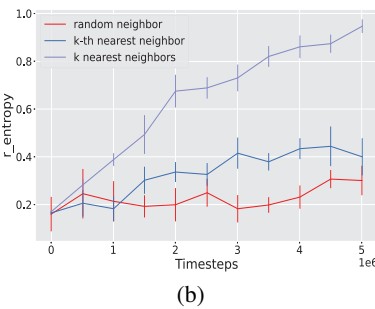

(a)                                                              (b)

Figure 6: (a) Performance comparisons of our method against different variants in the scenarios of SMAC. (b) Different kinds of intrinsic rewards in the corridor scenario.

## 4.3 ABLATION STUDY

We perform several ablation studies to examine the contribution of each component in our method. To evaluate the contribution of the autoregressive model used for learning trajectory representations, we create a variant that omits the autoregressive model, relying solely on the observation encoder. To evaluate the importance of contrastive representation learning, we develop a variant where trajectories are encoded by an encoder with fixed randomly initialized parameters. To test the effectiveness of the identity representations introduced in our method, we design two variants: the first employs the vanilla contrastive learning that directly contrasts trajectory samples against each other rather than using identity representations as medium variables to contrast different trajectory samples; the second replaces the learnable identity representations with fixed agent identities, such as one-hot vectors, in the contrastive learning loss. Additionally, we introduce a variant that learns trajectory representations through predicting the agent identities from trajectories in a supervised manner. Finally, to test the trajectory entropy, we develop two variants that respectively utilize randomly selected neighbors and the $k$-th nearest neighbor in the trajectory entropy.

We test these variants in three SMAC scenarios: 3s5z (easy), 2c_vs_64zg (hard), and corridor (super hard). The performance results are presented in Figure 6a. Employing the $k$-th nearest neighbor in the trajectory entropy damages performance and introduces significant variance. We also observe a noticeable decline in performance when using randomly selected neighbors. However, both variants still receive higher win rates than QMIX, demonstrating the robustness of our representation learning method. As shown in Figure 6b, employing $k$ nearest neighbors in the trajectory entropy provides more efficient intrinsic rewards compared to the other two methods, promoting thorough exploration.

Employing a fixed encoder for trajectory encoding results in suboptimal performance, demonstrating the importance of contrastive representation learning. This performance degradation occurs because the representations encoded by a fixed encoder provide inefficient intrinsic rewards for agent exploration. Compared to contrastive representation learning, the representations learned via predicting agent identities from trajectories lead to a significant performance decline. This decline occurs because representations guided by fixed agent identities result in strong mutual dependence, hindering efficient exploration. Using fixed agent identities in contrastive learning loss also leads to poor performance. Simply using the vanilla contrastive learning without identity representations results in significant performance drop. These variants demonstrate the importance of our contrastive learning loss using identity representations to learn distinguishable trajectory representations. We further provide the trajectory representations learned by different variants in Figure 7. We note that these variants do not necessarily learn distinguishable trajectory representations for the entropy maximization to work.

Furthermore, removing the autoregressive model results in performance comparable to our method in the 3s5z and 2c_vs_64zg scenarios, but it causes a significant performance decline in the super hard corridor scenario. This suggests that utilizing the autoregressive model for learning trajectory representations enhances robustness, particularly in more challenging multi-agent tasks.

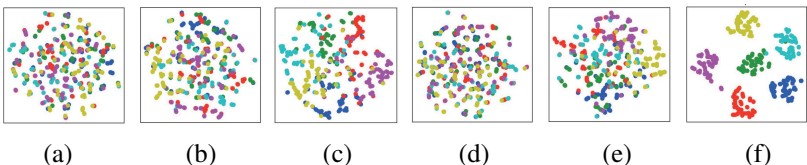

|  |  |  |  |  |  |
|---|---|---|---|---|---|
| (a) | (b) | (c) | (d) | (e) | (f) |

Figure 7: T-SNE plots for different agents' trajectory representations learned by different variants of TEE ((a) QMIX (b) TEE w/ fixed encoder (c) TEE w/ identity prediction (d) TEE w/ fixed agent identity (e) TEE w/ vanilla contrastive learning (f) TEE), in the corridor scenario of SMAC.

## 5 RELATED WORKS

In MARL settings, fostering diversity aims to encourage agents to visit diverse trajectories, resulting to varied policies among agents. SVO (McKee et al., 2020) uses social value orientation to address multi-agent social dilemmas by introducing intrinsic rewards that incentivize diverse policy learning. RODE (Wang et al., 2020c) enhances diversity by assigning agents to specific roles with distinct actions, though it may struggle in scenarios with continuous or large action spaces. MAVEN (Mahajan et al., 2019) adopts a value-based approach, conditioning agents' joint behaviors on a shared latent variable, controlled by a hierarchical policy, through maximizing mutual information. EOI (Jiang and Lu, 2021) trains a probabilistic classifier to predict the agent identities' probability distribution based on their observations, using the correct predictions as intrinsic rewards for policy training. CDS (Li et al., 2021) encourages diversity by optimizing mutual information through lower bounds based on the Boltzmann softmax distribution and variational inference. PMIC (Li et al., 2022) aims to foster the learning of superior policies by maximizing mutual information related to effective cooperative behaviors while minimizing it for less effective ones. LIPO (Charakorn et al., 2023) uses policy compatibility to develop diverse policies and further introduces variations in each agent's policy by maximizing mutual information. FoX (Jo et al., 2024) promotes formation-based exploration, fostering agents' understanding of their formations by encouraging them to explore diverse formations. Despite their achievements, these methods often overemphasize the dependence between agent identity and trajectories or formations, leading agents to repeatedly visit similar observations, which can limit their exploration of new possibilities. We refer the reader to extensive related works about entropy maximization in Appendix.

## 6 LIMITATIONS AND FUTURE WORKS

In this work, we simply use the distances between trajectory representations of different agents to measure the policy differences among agents. It could be an interesting direction to develop a more efficient policy difference measurement that could be used in the trajectory entropy estimator to significantly improve the performance of our method.

## 7 CONCLUSION

We propose a novel multi-agent exploration method to solve the limitation of homogeneous behaviors of agents caused by parameter sharing. Our method introduces an intrinsic reward based on a particle-based entropy estimator to maximize trajectory entropy in a contrastive representation space, which promotes efficient exploration. We test our method in a variety of challenging multi-agent tasks. The experimental results demonstrate the outperformance of our method compared to existing state-of-the-art methods.

### ACKNOWLEDGMENTS

This work was supported in part by the Fundamental Research Funds for the Central Universities (Grant No. NS2024055), in part by National Natural Science Foundation of China ( 62061146002), and in part by Natural Science Foundation of Jiangsu Province (Grant No. BK20222012 ).

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

## A  RELATED WORKS ABOUT ENTROPY MAXIMIZATION

**Entropy Maximization**: Entropy maximization has been adopted in various RL works to effectively promote state exploration. RE3 Seo et al. (2021) aims to improve sample efficiency by facilitating exploration. It achieves this by transforming high-dimensional observations into a compact, low-dimensional representation space using a fixed encoder, then employing an entropy estimator to assess state entropy within the representation space. Unlike RE3, we use contrastive learning to create a contrastive representation space, which includes more relevant information. APT Liu and Abbeel (2021) introduces a pre-training method that maximizes state entropy for sufficient exploration in unknown areas in the task-agnostic environment, utilizing an entropy-based intrinsic reward to train the agent's policy in a reward-free setting. ProtoRL Yarats et al. (2021) learns representations via prototypes that summarize the agent's exploration experience, which not only generalize across tasks but also accelerate exploration efficiently. Drawing inspiration from these approaches, our method in multi-agent settings promotes the diversity among agents by maximizing trajectory entropy in a contrastive representation space, thereby fostering both efficient exploration and effective collaboration.

## B  LIMITATION OF MUTUAL INFORMATION-BASED MULTI-AGENT EXPLORATION

In this section, we analyze the limitations of existing mutual information-based exploration methods from a theoretical perspective, particularly focusing on how agents tend to revisit familiar trajectories rather than exploring new ones. We present the reward functions associated with exploring both familiar and new trajectories. The theoretical results indicate that agents receive higher rewards when revisiting known trajectories compared to when they explore new ones.

The mutual information between the trajectory $\tau$ and agent identity $i$ is defined as:

$$\begin{aligned} I(i;\tau) &= \mathbb{E}_{i,\tau}[\log p(i \mid \tau)] - \mathbb{E}_i[\log p(i)] \\ &\geq \mathbb{E}_{i,\tau}\left[\log q_\theta(i \mid \tau)\right] - \mathbb{E}_i[\log p(i)] \end{aligned} \tag{10}$$

where the unknown posterior distribution $p(i \mid \tau)$ is approximated by a variational distribution $q_\theta(i \mid \tau)$. We parameterize $q_\theta(i \mid \tau)$ with $\theta$ and update $\theta$ to maximize the likelihood of $(i, \tau)$-tuples stored in the replay buffer. Prior works maximize mutual information by using the variational lower bound as an intrinsic reward:

$$\begin{aligned} r\left(\tau, i'\right) &= \log q_\theta\left(i' \mid \tau\right) - \log p\left(i'\right) \\ &= \log q_\theta\left(i' \mid \tau\right) + \log |A| \end{aligned} \tag{11}$$

where $p(i)$ is assumed to be a uniform distribution, so $-\log p\left(i'\right) = \log |A|$, where $|A|$ is the number of agents. We assume access to a perfect distribution $q_\theta(i \mid \tau)$, ensuring that $\sum_{a=1}^{|A|} q_\theta\left(i_a \mid \tau\right) = 1$.

**Intrinsic reward for familiar trajectories** The intrinsic reward function encourages agents to visit familiar trajectories $\tau$ where $q_\theta\left(i' \mid \tau\right) \to 1$. As a result:

$$r_{\text{known}} = \log 1 + \log |A| = \log |A|. \tag{12}$$

**Intrinsic reward for new trajectories** When agents visit new trajectories, where $q_\theta\left(i' \mid \tau\right)$ is unknown, we assign a null probability to unseen trajectories by adding a 'background' class to the model. The penalty agents receive when visiting these unseen trajectories is:

$$r_{\text{new}} = \lim_{q_\theta(i'|\tau)\to 0} \log q_\theta\left(i' \mid \tau\right) + \log |A| = -\infty \tag{13}$$

This analysis shows that as the distribution $q_\theta\left(i' \mid \tau\right)$ converges, agents are incentivized to revisit known trajectories, where they can achieve much higher rewards compared to exploring new, unfamiliar trajectories.

## C  THE TD LOSS OF QMIX

QMIX trains the policies of agents jointly by optimizing $Q_{tot}$ using the TD loss, defined as:

$$\mathcal{L}_{TD}^{QMIX} = \sum_{i=1}^{b} \left[ \left( r + \gamma \max_{\mathbf{u}_{t+1}} \bar{Q}_{tot}\left(s_{t+1}, \mathbf{u}_{t+1}\right) - Q_{tot}(s_t, \mathbf{u}_t) \right)^2 \right] \tag{14}$$

where $\bar{Q}_{tot}$ represents the target network, $b$ is the batch size of transition samples, and $\mathcal{D}$ is the replay buffer storing trajectory samples. $r$ denotes the global reward shared by all agents. Notably, since the policies of all agents are trained jointly by minimizing the TD loss, it is not possible to directly incorporate each agent's intrinsic reward $r_{entropy}^a$ into the global reward $r$ to independently train each agent's policy. This limitation necessitates the addition of an intrinsic utility network, $Q_a^{entropy}$, which is specifically designed to optimize the intrinsic reward $r_{entropy}^a$.

## D  THE IMPLEMENTATION OF TEE WITH POLICY GRADIENT METHODS

In our paper, we integrated our method with the value-based method QMIX. Here, we demonstrate how to integrate our proposed TEE with policy gradient methods. Specifically, we integrate TEE with MAPPO, a state-of-the-art policy-based MARL algorithm measured by SMAC. MAPPO involves training an actor network and a critic network shared among all agents. Because each agent trains its own critic, we can thus adopt a shaped reward, $r_{env} + \alpha r_{entropy}^a$ (where $r_{env}$ is the environmental reward and $r_{entropy}^a$ is the intrinsic reward generated by our method), when computing the reward-to-go $\hat{R}$ to train each agent's critic network. The other components of MAPPO remain unchanged. We conduct experiments on Pac-Men, SMAC, and SMACv2 to evaluate the performance of TEE+MAPPO. The results, presented in Table 1, show that TEE+MAPPO outperforms the baselines significantly.

## E  PYTORCH-STYLE PSEUDOCODE FOR TEE

The pytorch-style pseudocode for TEE is provided in Algorithm 1.

## F  ENVIRONMENTAL DETAILS AND ADDITIONAL EXPERIMENTAL RESULTS

In Pac-Men, we initialize four agents positioned in the central room of a maze. Each agent can only observe a 4×4 grid around them. There are some randomly initialized dots distributed in each edge room. The goal of the agent is to collect as many dots as possible in each edge room. We set different lengths for the paths to investigate the exploration of different MARL algorithms. Specifically, for the downward, left, right, and upward paths, the path lengths are 3, 6, 6, and 10, respectively. Only one path is within the agent's observation scope. The dots in each room will refresh when all of them are eaten by agents. The environmental reward received by the agent equals the total number of dots eaten in each time step.

SMAC benchmark is a set of cooperative tasks built on StarCraft II, aimed at evaluating the effectiveness of various Multi-Agent Reinforcement Learning (MARL) algorithms. The agent-level control in SMAC is achieved through the Machine Learning APIs provided by both StarCraft II and

---

**Algorithm 1:** PyTorch-style pseudocode for TEE

```
# batch:  collected trajectories
# |A|:  number of agents
# H: dimension of the identity representation
identity_representations = nn.Parameter(th.randn(|A|, H))
def TEE(batch):
   ctr_out = []
   for t in range(batch.seq_length):
      z_embedding = encoder(batch["obs"][:, t])
      c_embedding, hidden_states =
       autoregressive_model(z_embedding, hidden_states)
      ctr_out.append(c_embedding)
   ctr_out = th.stack(ctr_out, dim=1) # Concat trajectory
    representations over time
   trajectory_representation_loss =
    contrastive_loss(identity_representations, c_embedding)
   optimizer.zero_grad()
   trajectory_representation_loss.backward()
   optimizer.step()
   for t in range(batch.seq_length-1):
      z_embedding = encoder(batch["obs"][:, t+1])
      c_embedding, hidden_states =
       autoregressive_model(z_embedding, hidden_states)
      intrinsic_reward = entropy_estimator(c_embedding)
      intrinsic_rewards.append(intrinsic_reward)
   intrinsic_rewards = torch.stack(intrinsic_rewards, dim=1) #
    Concat intrinsic rewards over time
   return intrinsic_rewards
```

---

DeepMind's PySC2. Each task features a combat scenario with two armies: one controlled by allied RL agents and the other by a non-learning game AI. The game is over when all units of any army die or a predefined time limit is reached. The objective for the allied agents is to learn a policy that maximizes the game's win rate. To achieve this, agents must learn a sequence of actions to collaborate with allies in defeating enemy forces. An illustrative example of such collaboration is the mastery of kiting skills, where agents form formations based on their armor types, compelling enemy units to pursue while maintaining a safe distance to minimize damage. The SC2.4.10 version of StarCraft II is used, and it's important to note that performance comparisons between different versions are not applicable. We conduct experiments on six scenarios including 3s5z, 2c_vs_64zg, 7sz, 6h_vs_8z, corridor, and 3s5z_vs_3s6z with various difficulty levels.

SMAC has notable drawbacks due to the lack of stochasticity, and to address this issue, SMACv2 proposes some changes: introducing random team compositions, and random start positions. These adjustments aim to increase stochasticity to efficiently test the exploration of MARL algorithms. We conduct experiments on three scenarios of SMACv2 including terran_5_vs_5, protoss_5_vs_5, and zerg_5_vs_5. SMACv2 employs three unit types for each race. Units are algorithmically generated in teams, with each unit type having a fixed probability, consistent at both test and train times. The unit types of allied agents in these scenarios are identical to those of enemies. At the beginning of each episode, the allied agents are randomly spawned in the map with reflect or surround style.

We show the average returns of all algorithms in Pac-Men, SMAC, and SMACv2 with standard deviation over five random seeds in Table 1. The results demonstrate the significant outperformance of our method compared to baselines.

## G  ENVIRONMENTAL DETAILS AND EXPERIMENTAL RESULTS

In Pac-Men, four agents are initialized in the central room of a maze. Each agent has a limited observation range, confined to a $4 \times 4$ grid around them. Dots are randomly distributed in the edge

rooms of the maze, and the agents' goal is to collect as many dots as possible from these rooms. To examine the exploration capabilities of different MARL algorithms, we set varying path lengths: 3, 6, 6, and 10 steps for the downward, left, right, and upward paths, respectively. Only the downward path falls within the agents' observation range. The dots in each room respawn after all have been collected by the agents. The environmental reward each agent receives corresponds to the number of dots collected in each time step.

The SMAC benchmark consists of a set of cooperative tasks built on StarCraft II, designed to evaluate the effectiveness of various MARL algorithms. In SMAC, agent-level control is facilitated through the Machine Learning APIs provided by both StarCraft II and DeepMind's PySC2. Each task presents a combat scenario involving two armies: one controlled by allied RL agents and the other by a non-learning game AI. The game ends when all units of one army are eliminated or when a predefined time limit is reached. The allied agents aim to learn policies that maximize the win rate by collaborating effectively to defeat enemy forces. An example of such collaboration is mastering kiting skills, where agents form formations based on their armor types, forcing enemy units to pursue them while maintaining a safe distance to minimize damage. The experiments utilize version SC2.4.10 of StarCraft II, and it's important to note that performance comparisons across different versions are not applicable. We conduct experiments on six scenarios with varying difficulty levels: 3s5z, 2c_vs_64zg, 7sz, 6h_vs_8z, corridor, 3s5z_vs_3s6z, MMM2, 10m_vs_11m, and 27m_vs_30m.

SMAC has some limitations due to its lack of stochasticity. To address this, SMACv2 introduces modifications such as random team compositions and random start positions. These changes aim to enhance stochasticity, making it a more effective benchmark for evaluating the exploration capabilities of MARL algorithms. We conduct experiments on three SMACv2 scenarios: terran_5_vs_5, protoss_5_vs_5, and zerg_5_vs_5. SMACv2 sets three unit types for each race, with teams being algorithmically generated based on a fixed probability for each unit type, consistent during both testing and training phases. The allied agents' unit types are identical to those of the enemies. At the start of each episode, the allied agents are randomly spawned on the map in either a reflective or surround formation.

We also test our method in two scenarios of Google Research Football (GRF). GRF provides a challenging, physics-based environment that simulates a football game where agents need to learn strategic planning, coordination, and precise timing to succeed. The left side players (except the goalkeeper) as agents are trained to learn cooperative policies. The right side palyers are controlled by the game engine. The agents operate within a discrete action space of 19 options, including moving in eight directions, sliding, shooting, and passing. Agent observations include the positions and movement directions of the controlled agent, other agents, and the ball.

We present the average returns of all algorithms in Pac-Men, SMAC, SMACv2, and GRF along with the standard deviation over five random seeds, in Table 1. The results demonstrate that our method significantly outperforms baseline methods.

Table 1: Average returns of all algorithms in Pac-Men, SMAC, SMACv2, and GRF. $\pm$ denotes the standard deviation over five random seeds.

| Method | Pac-Men | SMAC | | | | | | | | | SMACv2 | | | GRF | | |
|---|---|---|---|---|---|---|---|---|---|---|---|---|---|---|---|---|
| | | 3s5z | 2c_vs_64zg | 7sz | 6h_vs_8z | corridor | 3s5z_vs_3s6z | MMM2 | 10m_vs_11m | 27m_vs_30m | terran_5_vs_5 | protoss_5_vs_5 | zerg_5_vs_5 | academy_3_vs_1_with_keeper | academy_4_vs_2_with_keeper | academy_counter_attack_hard |
| QMIX | 0.21±0.04 | 0.72±0.13 | 0.85±0.08 | 0.17±0.02 | 0.23±0.03 | 0.57±0.07 | 0.36±0.12 | 0.27±0.06 | 0.57±0.04 | 0.43±0.07 | 0.68±0.03 | 0.53±0.05 | 0.41±0.04 | 0.23±0.05 | 0.13±0.09 | 0.17±0.03 |
| MAPPO | 0.49±0.03 | 0.81±0.05 | 0.83±0.04 | 0.52±0.06 | 0.53±0.03 | 0.62±0.05 | 0.57±0.08 | 0.46±0.03 | 0.39±0.05 | 0.43±0.05 | 0.52±0.04 | 0.47±0.03 | 0.37±0.03 | 0.31±0.09 | 0.18±0.09 | 0.23±0.07 |
| MAVEN | 0.32±0.06 | 0.51±0.21 | 0.72±0.06 | 0.00±0.00 | 0.42±0.04 | 0.36±0.08 | 0.18±0.15 | 0.43±0.11 | 0.62±0.08 | 0.53±0.09 | 0.58±0.04 | 0.31±0.05 | 0.29±0.03 | 0.18±0.06 | 0.08±0.06 | 0.13±0.09 |
| EOI | 0.41±0.05 | 0.87±0.07 | 0.83±0.02 | 0.37±0.03 | 0.08±0.03 | 0.25±0.11 | 0.42±0.13 | 0.39±0.08 | 0.72±0.03 | 0.64±0.06 | 0.65±0.05 | 0.42±0.03 | 0.47±0.04 | 0.17±0.05 | 0.05±0.03 | 0.07±0.03 |
| QTRAN | 0.28±0.08 | 0.21±0.19 | 0.75±0.05 | 0.00±0.00 | 0.02±0.02 | 0.08±0.07 | 0.02±0.01 | 0.13±0.05 | 0.43±0.03 | 0.21±0.07 | 0.42±0.02 | 0.40±0.04 | 0.25±0.02 | 0.25±0.03 | 0.13±0.08 | 0.11±0.05 |
| SCDS | 0.37±0.05 | 0.76±0.07 | 0.57±0.09 | 0.21±0.03 | 0.03±0.01 | 0.56±0.06 | 0.00±0.00 | 0.32±0.08 | 0.62±0.03 | 0.57±0.09 | 0.52±0.03 | 0.47±0.05 | 0.38±0.04 | 0.42±0.13 | 0.25±0.11 | 0.47±0.06 |
| PMIC | 0.34±0.03 | 0.82±0.03 | 0.79±0.05 | 0.58±0.02 | 0.51±0.05 | 0.37±0.03 | 0.18±0.06 | 0.19±0.05 | 0.43±0.07 | 0.62±0.06 | 0.47±0.03 | 0.36±0.02 | 0.42±0.02 | 0.23±0.08 | 0.11±0.07 | 0.16±0.07 |
| LIPO | 0.43±0.02 | 0.71±0.03 | 0.76±0.02 | 0.39±0.04 | 0.36±0.06 | 0.27±0.03 | 0.21±0.03 | 0.27±0.13 | 0.52±0.04 | 0.37±0.04 | 0.43±0.02 | 0.46±0.03 | 0.37±0.03 | 0.19±0.05 | 0.07±0.03 | 0.12±0.05 |
| FoX | 0.39±0.03 | 0.74±0.02 | 0.64±0.05 | 0.56±0.03 | 0.45±0.05 | 0.52±0.04 | 0.43±0.04 | 0.32±0.13 | 0.67±0.05 | 0.61±0.09 | 0.54±0.03 | 0.56±0.02 | 0.49±0.02 | 0.57±0.05 | 0.41±0.13 | 0.33±0.08 |
| TEE+QMIX | **0.83±0.03** | 0.95±0.02 | **0.96±0.03** | 0.83±0.04 | 0.85±0.03 | **0.90±0.03** | 0.87±0.04 | **0.85±0.03** | **0.93±0.05** | 0.89±0.05 | **0.96±0.02** | **0.95±0.03** | **0.87±0.03** | **0.79±0.13** | **0.63±0.16** | **0.71±0.09** |
| TEE+MAPPO | 0.80±0.04 | **0.97±0.02** | 0.94±0.04 | **0.90±0.03** | **0.87±0.04** | 0.85±0.03 | **0.94±0.03** | 0.76±0.13 | 0.90±0.06 | **0.94±0.06** | 0.87±0.02 | 0.89±0.02 | 0.84±0.05 | 0.68±0.08 | 0.61±0.12 | 0.57±0.06 |

## H    TRAINING DETAILS AND HYPERPARAMETERS

The overall trajectory encoder used in our method is composed of an encoder and an autoregressive model. For the encoder, we utilize two MLP layers with a hidden size of 64, followed by batch normalization. The autoregressive model is implemented using a GRU unit. We use randomly initialized learnable vectors as identity representations, which have the same dimensions as the trajectory representations. We integrate TEE with QMIX by adding an intrinsic utility network, which

is composed of a two-layer MLP with a hidden size of 64, optimized to maximize the total intrinsic rewards. All other components remain identical to the standard QMIX.

To ensure a fair comparison, the policy networks for all agents are constructed using Deep Recurrent Q-Networks. At each time step, an agent's policy network processes a local observation as well as the last step action through a fully connected hidden layer, a GRU unit, and a final fully connected layer that generates $|U|$ outputs, corresponding to the available actions. We set the evaluation interval to 10K steps followed by 32 test episodes. We run all methods for 5 million steps. In SMAC and SMACv2, target networks are updated using hard updates every 200 episodes, whereas in Pac-Men, soft updates with a momentum of 0.01 are employed. The hyperparameters for TEE and baseline methods in Pac-Men, SMAC, and SMACv2 are detailed in Table 2. Parameter sharing is applied across all methods to enable agents to make action decisions. For generality, we report both the mean and standard deviation of performance results, averaged over five random seeds. To ensure a fair comparison, consistent hyperparameters are used across different methods. The replay buffer size is set to 5K. We implemented our method using NumPy and PyTorch, and all experiments are conducted on a single NVIDIA GeForce RTX 4090 GPU.

Table 2: Hyperparameters

| | Pac-Men | SMAC | | SMACv2 |
|---|---|---|---|---|
| hidden dimension | 64 | | 128 | |
| learning rate | 0.0003 | | 0.005 | |
| optimizer | | | Adam | |
| target update | 0.01(soft) | | 200(hard) | |
| batch size | 32 | | 64 | |
| $\beta$ | 0.05 | 0.05 for 3s5z, 2c_vs_64zg, 8m, 5m_vs_6m, 8m_vs_9m, and 10m_vs_11m, 0.02 for 7sz, 6h_vs_8z, corridor, and 3s5z_vs_3s6z | | 0.05 |
| $\alpha$ | 0.02 | 0.01 for 3s5z, 2c_vs_64zg, 8m, 5m_vs_6m, 8m_vs_9m, and 10m_vs_11m, 0.02 for 7sz, 6h_vs_8z, corridor, and 3s5z_vs_3s6z | | 0.01 |
| $k$ | 3 | 8 for 10m_vs_11m, 5 for 3s5z, 7sz, corridor, 3s5z_vs_3s6z, 8m, 5m_vs_6m, and 8m_vs_9m, 1 for 2c_vs_64zg, 4 for 6h_vs_8z | | 4 for terran_5_vs_5, protoss_5_vs_5, and zerg_5_vs_5, 7 for terran_10_vs_10, 12 for terran_15_vs_15, 18 for terran_20_vs_20 |
| epsilon anneal time | 200,000 | 200,000 for 3s5z, 2c_vs_64zg, 8m, 5m_vs_6m, 8m_vs_9m, and 10m_vs_11m, 500,000 for 7sz, 6h_vs_8z, corridor, and 3s5z_vs_3s6z | | 500,000 |

## I EVALUATIONS OF TEE WITH DIFFERENT VALUES OF $k$

The values of $k$ used in different experimental environments are listed in Table 2. To investigate whether the performance of our method is strongly sensitive to $k$, we show the performance of our method with different values of $k$ in the terran_5_vs_5 (including 5 agents) and terran_20_vs_20 (including 20 agents) scenarios in Table 3. We note that different values of $k$ only lead to small differences in performance in both scenarios. The performance of our method remains robust across different values of $k$.

## J EVALUATION OF TEE IN SCENARIOS REQUIRING HOMOGENEOUS BEHAVIOR

While our proposed TEE succeeds in promoting diversity among agents to encourage exploration, there are situations where agents may benefit from behaving uniformly, especially in simpler scenarios. For instance, allied agents might employ the same tactic, such as simultaneously firing at a single enemy to quickly eliminate it. To demonstrate our method's ability to learn such behaviors, we test it in four homogeneous SMAC scenarios where the focus fire tactic is advantageous. The results, shown in Table 4, indicate that our method consistently outperforms QMIX in all scenarios, confirming that it can support uniform behaviors when they lead to higher environmental rewards. This phenomenon demonstrates that our method can efficiently balance exploration and exploitation, leading to optimal cooperative behaviors.

Table 3: Performance of our method with different values of $k$

| Methods | terran_5_vs_5 | | | | terran_20_vs_20 | | | | |
|---|---|---|---|---|---|---|---|---|---|
| | k=1 | k=2 | k=3 | k=4 | k=1 | k=4 | k=10 | k=15 | k=18 |
| TEE+QMIX | 0.92 ±0.03 | 0.90 ±0.03 | 0.93 ±0.04 | 0.96 ±0.02 | 0.85 ±0.03 | 0.86 ±0.04 | 0.83 ±0.04 | 0.88 ±0.05 | 0.89 ±0.03 |

Table 4: Performance of our method and QMIX in homogeneous scenarios.

| Method | 8m | 5m_vs_6m | 8m_vs_9m | 10m_vs_11m |
|---|---|---|---|---|
| TEE+QMIX | 0.95±0.03 | 0.92±0.04 | 0.94±0.03 | 0.91±0.06 |
| QMIX | 0.87±0.03 | 0.65±0.04 | 0.58±0.05 | 0.43±0.04 |

Table 5: Performance comparisons of our method against QMIX with different values of $\epsilon$

| Methods | corridor | 3s5z_vs_3s6z |
|---|---|---|
| Trajectory entropy maximization (Ours) | 0.90 ±0.03 | 0.87 ±0.04 |
| $\epsilon = 0.05$ | 0.57 ±0.07 | 0.36 ±0.12 |
| $\epsilon = 0.08$ | 0.61 ±0.04 | 0.39 ±0.09 |
| $\epsilon = 0.12$ | 0.65 ±0.06 | 0.45 ±0.14 |

## K   COMPARISON WITH $\epsilon$-GREEDY

$\epsilon$-greedy is a widely used exploration technique in many RL approaches. Increasing the value of $\epsilon$ generally promotes better exploration. In this section, we compare our entropy maximization method with $\epsilon$-greedy to demonstrate the effectiveness of our method in promoting exploration in the domain of MARL. To achieve this, we set the $\epsilon$ values to 0.05, 0.08, and 0.12 for QMIX, respectively. We test different values of $\epsilon$ in the super hard scenarios corridor and 3s5z_vs_3s6z. The results are shown in Table 5. Our entropy maximization method is more efficient than increasing the values of $\epsilon$ to encourage effective exploration in multi-agent environments. We note that increasing the values of $\epsilon$ does not improve the performance significantly. In multi-agent settings, larger $\epsilon$ values only increase the stochasticity in action selections of a single agent and ignore the diversity or distinguishability among agents as they do not consider the trajectories of other agents, thus leading to inefficient exploration.

## L   SCALABILITY

Many MARL methods suffer from poor scalability, i.e., the agent performance decreases significantly as the number of agents increases. This occurs because the state-action space expands exponentially as the number of agents increases, making efficient exploration crucial. In this section, we evaluate the scalability of our method across four SMACv2 scenarios with increasing numbers of agents: terran_5_vs_5, terran_10_vs_10, terran_15_vs_15, and terran_20_vs_20. The results, presented in Table 6, show that our method substantially outperforms QMIX across all scenarios. QMIX faces challenges in scalability and struggles to achieve satisfactory performance due to insufficient exploration. In contrast, our method shows strong scalability, demonstrating that maximizing trajectory entropy enables efficient exploration.

## M   VISUALIZATIONS

We present some visualization examples of the diverse policies learned by our method, as shown in Figure 8, which emerge in the extremely challenging scenarios (6h_vs_8z, corridor, and 3s5z_vs_3s6z). For instance, in the 6h_vs_8z scenario, one agent moves in the opposite direction of the team to cover other agents. Most of the enemies are then drawn to the agent's movements. The agent continues kiting the pursuing enemies, drawing the majority of their fire. Meanwhile, the remaining enemies are surrounded by the other agents. This strategy allows agents to cooperatively distribute enemies' attacks. In contrast, if all agents behaved similarly and rushed toward the enemies, they would be

Table 6: Performance of our method and QMIX in scenarios of SMACv2 with different number of agents

| Method | terran_5_vs_5 | terran_10_vs_10 | terran_15_vs_15 | terran_20_vs_20 |
|---|---|---|---|---|
| TEE+QMIX | 0.96±0.02 | 0.95 ±0.03 | 0.92 ±0.04 | 0.89 ±0.03 |
| QMIX | 0.68±0.03 | 0.39±0.04 | 0.24 ±0.06 | 0.11±0.05 |

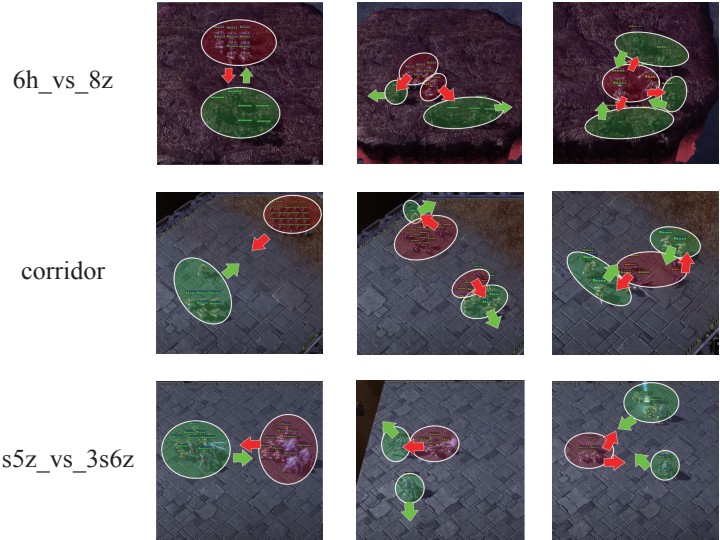

Figure 8: Visualization examples of diverse policies emerging in 6h_vs_8z (top), corridor (medium), and 3s5z_vs_3s6z (bottom) from initial (left) to final (right). Green and red shadows represent agents and enemies, respectively. Green and red arrows represent the moving directions of agents and enemies, respectively.

quickly defeated by the stronger opponents. Similar diverse policies are observed in the other two scenarios. These results highlight the effectiveness of our method in learning diverse policies, helping agents cooperatively defeat enemies.

We also provide additional visitation heatmaps for TEE+QMIX and the baseline methods in SMACv2 scenarios, as shown in Figures 9 and 10. These heatmaps illustrate that our proposed TEE achieves more efficient exploration compared to the baselines. The mutual information-based baselines such as MAVEN, EOI, and SCDS do not sufficiently explore the environment, making them less effective at defeating enemies that appear randomly on the map. While the mutual information objective encourages multi-agent diversity, it can also impede effective exploration. In contrast, our proposed TEE leverages the trajectory entropy maximization objective, which drives agents to adopt exploratory policies without being constrained by mutual dependence.

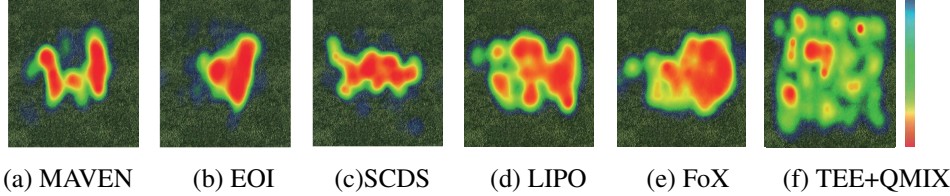

(a) MAVEN     (b) EOI     (c)SCDS     (d) LIPO     (e) FoX     (f) TEE+QMIX

Figure 9: Visitation heatmaps of different algorithms in the protoss_5_vs_5 scenario.

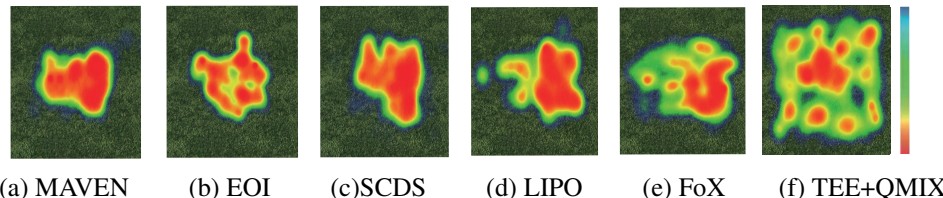

(a) MAVEN    (b) EOI    (c)SCDS    (d) LIPO    (e) FoX    (f) TEE+QMIX

Figure 10: Visitation heatmaps of different algorithms in the zerg_5_vs_5 scenario.

