# OpenReview forum: "Toward Efficient Multi-Agent Exploration With Trajectory Entropy Maximization"
_ICLR.cc/2025/Conference — ICLR 2025 Poster_

### Official Review · Reviewer_6aVA · 2024-10-31

**Soundness:** 2
**Presentation:** 3
**Contribution:** 2
**Rating:** 3
**Confidence:** 5

**Summary:**

This paper aims to enhance exploration in MARL by maximizing trajectory entropy (TEE). The authors propose encoding trajectories into low-dimensional vector representations through contrastive learning, calculating particle-based entropy on these representations, and maximizing it as an intrinsic reward. This method is incorporated into the QMIX algorithm and tested in grid-world and SMAC/SMAC-v2 environments, with experimental results indicating that TEE outperforms other QMIX-based exploration techniques.

Although the paper is well-structured with a coherent logic flow, I have concerns regarding its unclear motivation and potentially problematic experimental results. It shares a common shortcoming with MARL research from a few years ago. I recommend a rejection.

**Strengths:**

* The paper is well-written with a logical flow.

* The authors include an extensive set of experimental baselines; however, some important baselines are missing, as I elaborate in the next section.

**Weaknesses:**

* **Policy-Gradient-Based Algorithms**. This paper does not address any policy-gradient-based (PG) MARL algorithms, raising two issues. First, it's unclear whether insufficient exploration is a significant problem for PG algorithms as well. Could you also include policy gradient algorithms like MAPPO [1] and HAPPO [2] in the Pac-Men experiments? At the very least, these works are relevant and should be cited. Second, policy gradient methods have proven competitive on benchmarks such as SMAC. Even if these algorithms aren't directly compared on the learning curve, their final performance should be used as a baseline so readers can see that your approach advances the MARL field. Omitting them weakens the paper's overall rigor.

[1] The Surprising Effectiveness of PPO in Cooperative, Multi-Agent Games, https://arxiv.org/abs/2103.01955

[2] TRUST REGION POLICY OPTIMISATION IN MULTI-AGENT REINFORCEMENT LEARNING, https://arxiv.org/pdf/2109.11251

* **Pac-Men Results Conflict.** Results in the Pac-Men environment appear to conflict with the published CDS paper [3]. The CDS paper shows this environment is solvable with 2 million timesteps, but in your results, it isn't. Please explain this discrepancy in detail. Additionally, while explaining your algorithm's success, you should analyze why baseline methods fail rather than providing vague reasons like "insufficient cooperation". Consider analyzing trajectories in the replay buffer, visualizing the Q-value heatmap, or conducting a more thorough breakdown.

[3] Celebrating Diversity in Shared Multi-Agent Reinforcement Learning, https://arxiv.org/pdf/2106.02195

* **Insufficient Evaluation.** The evaluation approach needs refinement. First, only six maps were selected in the SMAC environment, which raises concerns about potential cherry-picking. Second, Figure 3 shows a significant divergence from the curves in the CDS paper. Although the authors suggest this is due to differences in StarCraft engine versions, it would be best to match the version used in the CDS paper to ensure fair performance comparisons. Third, recent work has introduced more reliable MARL evaluation protocols [4,5], recommending evaluation across multiple domains (e.g., GRF, Melting Pot, SMAC) and reporting IQM scores/curves for each domain. Adopting these standards would improve the reliability of your evaluations.

[4] JaxMARL: Multi-Agent RL Environments in JAX, https://arxiv.org/abs/2311.10090

[5] Towards a Standardised Performance Evaluation Protocol for Cooperative MARL, https://arxiv.org/abs/2209.10485

**Questions:**

* Is the trajectory representation learned on-the-fly from the replay buffer, or is it pre-trained?

* Can your exploration method be applied to policy-gradient methods?

* Why did you choose QMIX as the base algorithm instead of QPLEX[6], which is more advanced and theoretically general?

[6] QPLEX: Duplex Dueling Multi-Agent Q-Learning, https://arxiv.org/abs/2008.01062

---

> ### Author Response · Authors · 2024-11-24
>
> Thank you for taking the time to review our paper. We clarify your concerns and problems below:
>
> Weakness 1: Policy-Gradient-Based Algorithms
>
> Due to the page limit, we present the details of the integration of our method with MAPPO in Appendix D, which is referenced in the last paragraph of Section 3.3. We compare TEE+MAPPO with MAPPO in four benchmarks including Pac-Men, SMAC, SMACv2, and GRF. For a fair comparison, we use consistent network structures and hyperparameters. The results are shown in Table 1 in the Appendix. Compared to MAPPO, TEE+MAPPO maintains its outperformance across all tasks. We have added the citations of MAPPO and HAPPO in the third paragraph of Section 1.
>
> Weakness 2: Pac-Men Results Conflict
>
> The settings of Pac-Men used in our method, depicted in Appendix F, are different from those used in the CDS paper. First, each agent can only observe a 4×4 grid around them in our Pac-Men instead of the 5×5 adopted in the CDS paper. Second, for the downward, left, right, and upward paths, the path lengths are 3, 6, 6, and 10 in our Pac-Men instead of 4, 8, 8, and 12 used in the CDS paper. Moreover, we use consistent hyperparameters and network structures for all baseline methods implemented in our paper to ensure a fair comparison. These may be different from those adopted in the CDS paper. Thus, these differences lead to different results achieved by CDS.  In both our paper and the CDS paper, we run the proposed methods for 5 million training steps in Pac-Men, as demonstrated by the training results shown in Figure 3 of the CDS paper, rather than the 2 million timesteps you mentioned.
>
> In our paper, we analyze why baseline methods fail by providing visitation heatmaps of different baseline methods in Figure 2(c)(d), 5, 9, and 10, which demonstrate insufficient state space exploration of baseline methods. We also compare the intrinsic reward of our method with that of mutual information-based baseline methods in Figure 2(e), highlighting why mutual information-based approaches are prone to overfitting. Moreover, we provide detailed discussions in our paper on why these baseline methods fail in each benchmark, such as the second paragraph of Section 4.1, and the third and the sixth paragraphs of Section 4.2.
>
> Weakness 3: Insufficient Evaluation
>
> For generality, we additionally test our method in three SMAC scenarios including MMM2, 10m\_vs\_11m, and 27m\_vs\_30m. The results are shown in Table 1 in our paper.
>
> We compare our method with CDS in SMAC scenarios, using the version mentioned in the CDS paper. We use consistent hyperparameters and the same network structure to ensure fairness. The evaluation results are shown below:
>
>
> |  Method   | 6h\_vs\_8z  |corridor  |3s5z\_vs\_3s6z  |
> |---------------------------------------------|-------------------|------------------|------------------|
> | CDS | 0.53 $\pm$ 0.06 |0.68 $\pm$ 0.11 |0.65 $\pm$ 0.09 |
> | TEE | 0.89 $\pm$ 0.05 |0.92 $\pm$ 0.08 |0.83 $\pm$ 0.06|
>
> With consistent hyperparameters, the same network structure, and the same SMAC version, our method outperforms CDS across all scenarios.
>
>
>
> To improve the reliability of evaluations, we test our method in three tasks of GRF. The final results are provided in Table 1. In all tasks of GRF, our method achieves superior performance compared to baseline methods.
>
> Q1: Is the trajectory representation learned on-the-fly from the replay buffer, or is it pre-trained?
>
> It is not pre-trained. We sample trajectories from the replay buffer to learn trajectory representations using contrastive learning.
>
> Q2: Can your exploration method be applied to policy-gradient methods?
>
> See Weakness 1.
>
> Q3: Why did you choose QMIX as the base algorithm instead of QPLEX[6], which is more advanced and theoretically general?
>
> QPLEX can be treated as a variant of QMIX, which incorporates a duplex dueling network architecture within the mixing network to enhance its representation capabilities. In our paper, we do not consider the performance improvement of QPLEX, due to optimizations in the network architecture of QMIX. We just need a more general, essential, simple, and reliable network architecture. Thus, we choose QMIX.
>
> We hope to receive your feedback soon and greatly appreciate the time you have taken to review our paper.

---

> > ### Comment · Reviewer_6aVA · 2024-11-26
> > **Thank you for the informative response**
> >
> > Thank you for the informative response. My concerns regarding policy gradient algorithms and the use of QMIX have been well addressed. However, I remain unconvinced about several aspects of the experiments.
> >
> > First, while I appreciate the authors' explanation of the Pac-Men environment, its design seems contrived. CDS can solve the task with environment configuration A but not configuration B. This raises the question of whether the Pac-Men environment is a proper benchmark for evaluating algorithm performance. For instance, TEE might also fail under another configuration C. The experimental results in their current form are not rigorous enough. I recommend conducting experiments across a broader range of configurations, such as searching grid sizes from 4x4 to 10x10 and increasing the path lengths in a similar manner. It is essential to identify the difficulty levels TEE and other baselines can solve. Such results would also be valuable for future algorithm development.
> >
> > Thank you for providing additional experimental results in the SMAC and GRF domains. However, I find the overall experimental evaluation not sufficiently reliable for publication. There are notable performance gaps compared to results reported in the CDS paper. For example, the win rate in the 6h vs 8z scenario is approximately 0.75 in the CDS paper but drops to 0.53 in your results. Have you communicated with the authors of CDS to confirm detailed experimental configurations? Is the discrepancy due to an inability to reproduce the results using the same configuration, or some other reason? Such details should be annotated in the paper to avoid confusion.
> >
> > Additionally, I recommend performing thorough comparisons on all maps using the old SC2 version employed in the CDS paper. Furthermore, interpreting the performance gap based solely on individual learning curves is challenging. I strongly recommend using tools like **RLiable** to provide aggregated and statistically robust results.

---

> > > ### Author Response · Authors · 2024-11-26
> > >
> > > Thank you for your response.
> > >
> > > We use Pac-Men to demonstrate the effectiveness of our method in learning diverse policies. We will evaluate our methd in different settings of Pac-Men and compare its performance with CDS. We will provide the results as soon as possible.
> > >
> > > As we discussed in our response, to ensure a fair comparison, we use consistent hyperparameters and the same network structures for all baseline methods. These hyperparameters and network structures may be different from those used in the CDS paper, which may lead to different experimental results. Moreover, we tried running the provided CDS code to reproduce the results; however, we did not achieve the same outcomes. Most of the results did not live up to what they presented in the paper. We are trying to communicate with the authors of CDS to confirm the results provided in their paper. We will test our method in the old version of SMAC employed in the CDS paper. Since it is a complicated work, we will provide the results as soon as possible.

---

> > > ### Author Response · Authors · 2024-11-26
> > > **Clarifications on the settings of Pac-Men in our paper**
> > >
> > > We can adjust the settings of Pac-Men, particularly the sight range of the agents, to make the environment more challenging. In our paper, we introduce a more challenging configuration of Pac-Men compared to that used in the CDS paper. Specifically, we reduce the sight range of Pac-Men from 5x5, as used in the CDS paper, to 4x4. This change was made because we found that our method performed satisfactorily with larger sight ranges. A more challenging environment is necessary to effectively test the robustness of our method.

---

> > > ### Author Response · Authors · 2024-11-28
> > >
> > > To evaluate the robustness of our method, we test it in Pac-Men, using different sight ranges and path lengths.
> > >
> > > We first compare the performance of our method with CDS in Pac-Men with different sight ranges from 3x3 to 10x10. We keep the path lengths unchanged (downward: left: right: upward = 3: 6: 6:  10). For a fair comparison, we adopt consistent hyperparameters and the same network structures. The average returns are shown below:
> > >
> > > | Method | 3x3 | 4x4 |  5x5 |  6x6 | 7x7|  8x8 | 9x9 | 10x10 |
> > > |-------|-------|-------|-------|-------|-------|-------|-------|-------|
> > > | CDS | 25$\pm$7  | 37 $\pm$5 |79$\pm$8 | 86$\pm$4 |90$\pm$5 | 92$\pm$7|97$\pm$3 | 103$\pm$4 |
> > > | TEE | 76$\pm$5 | 83 $\pm$3 |91$\pm$6 | 95$\pm$5 |103$\pm$7| 108$\pm$9 |117$\pm$6 | 125$\pm$11|
> > >
> > > We observe that as the agent's sight range decreases, our method achieves more robust performance compared to CDS. The performance of CDS degrades significantly when the agent's sight range drops below 5x5.
> > >
> > > We next evaluate our method in Pac-Men, using different path lengths while keeping the sight range fixed ( 4x4 ). The average returns are shown below:
> > >
> > >
> > > | Method | 3 : 6 : 6:  10 | 5 : 8 : 8:  12 |  7 : 10 : 10 :  14 |
> > > |-------|-------|-------|-------|
> > > | CDS | 37 $\pm$5  | 23 $\pm$8 |15$\pm$7 |
> > > | TEE | 83 $\pm$3 | 79 $\pm$5 |76$\pm$9 |
> > >
> > > The results demonstrate the robustness of our method. The performance of CDS decreases significantly as the path length increases. Compared to mutual information-based method such as CDS, our method provides a more efficient exploration method by maximizing the trajectory entropy.
> > >
> > > In our paper, we use the same version of SMAC (SC2) to test all the methods to guarantee fairness. In the previous response, we have evaluated our method in three super hard scenarios of the old version SMAC used in the CDS paper. Next, we compare the performance of our method with CDS in the remaining three scenarios of SMAC used in the CDS paper. We use the same hyperparameters and network structures to ensure fairness.
> > >
> > > |  Method | MMM2  |5m\_vs\_ 6m  |3s\_vs\_5z  |
> > > |---------------------------------------------|-------------------|------------------|------------------|
> > > |CDS | 0.76 $\pm$ 0.05 |0.53 $\pm$ 0.06 |0.69 $\pm$ 0.08|
> > > |TEE | 0.94 $\pm$ 0.03 |0.87 $\pm$ 0.04 |0.91 $\pm$ 0.03 |
> > >
> > > Our method outperforms CDS across all scenarios, demonstrating the effectiveness of our method in promoting exploration. We note that our evaluation results of CDS are different from those presented in the CDS paper. Similar performance discrepancies can also be observed in many other previous works, such as [1], [2], and [3]. We believe this is because different hyperparameters and network structures produce different experimental results. To solve this issue, we recommend building a public MARL training framework with consistent hyperparameters and network structures to ensure a fair comparison.
> > >
> > > [1] Liu, Yuntao, et al. "Heterogeneous skill learning for multi-agent tasks." Advances in Neural Information Processing Systems 35 (2022): 37011-37023.
> > >
> > > [2] Liu, Zichuan, Yuanyang Zhu, and Chunlin Chen. "NA$^2$ Q: Neural Attention Additive Model for Interpretable Multi-Agent Q-Learning." International Conference on Machine Learning. PMLR, 2023.
> > >
> > > [3] Na, Hyungho, Yunkyeong Seo, and Il-chul Moon. "Efficient Episodic Memory Utilization of Cooperative Multi-Agent Reinforcement Learning." The Twelfth International Conference on Learning Representations.

---

> > > ### Author Response · Authors · 2024-12-01
> > >
> > > Thank you for your insightful suggestions. We look forward to your feedback.

---

> > > > ### Comment · Reviewer_6aVA · 2024-12-01
> > > >
> > > > Thank you for the response, but it does not fully address my concerns. While I appreciate the authors' additional results in the Pac-Men and SMAC environments, I still have doubts about the validity of the general evaluation protocol used in the experiments. The issue extends beyond discrepancies with the CDS baseline. Although the authors claim to have used consistent hyperparameters across baselines to ensure fair comparisons, why are these hyperparameters deemed appropriate or optimal? It is entirely possible that with proper tuning (e.g., using a larger batch size), some baselines could achieve significantly higher rewards. While I am not suggesting that the authors perform exhaustive hyperparameter tuning for every algorithm, one alternative could be to directly adopt results from the original papers when reproduction is not possible, rather than re-running the baselines with suboptimal settings that down-tune their performance.
> > > >
> > > > A related question is, how were the random seeds chosen? Were they selected systematically (e.g., seeds 1 through 5) or arbitrarily? Were the same seeds applied consistently across all baselines? These details are crucial for ensuring fair and reproducible comparisons.
> > > >
> > > > To further illustrate my point, consider the reported performance of MAPPO in Table 1 of this paper compared to the original paper ([1], Table 1). For the 2cvs64zg map, MAPPO achieves a reward of 0.97 in [1], but only 0.83 in this paper.
> > > > For the corridor map, MAPPO's reward is 0.94 in [1], yet only 0.62 in this paper. These discrepancies strongly suggest that the experimental setup may not reflect the true performance of the baselines. I recommend the authors to thoroughly re-examine their experiments, identify any inconsistencies, and provide clear explanations for these discrepancies in the paper. This process should prioritize scientific rigor over simply addressing a reviewer's concerns.
> > > >
> > > > [1] The Surprising Effectiveness of PPO in Cooperative Multi-Agent Games

---

> > > > > ### Author Response · Authors · 2024-12-03
> > > > >
> > > > > Directly adopting results from the original papers is unreasonable. The hyperparameters and network structures may vary across various papers. To ensure a fair comparison, in our work, we need to use consistent hyperparameters and network structures, as done in [1-3], which has been a evaluation standard that is widely adopted to test different methods. The reasons are as follows:
> > > > >
> > > > > First, hyperparameters and network structures significantly influence MARL performance. By standardizing them, we ensure that observed performance differences arise from the methods themselves rather than extraneous factors. The key contribution of a MARL paper typically lies in the algorithmic innovation. Keeping all other factors fixed highlights the method's true capabilities without being clouded by secondary influences like architecture design or parameter tuning.
> > > > >
> > > > > Second, when testing different algorithms, the goal is to understand their relative effectiveness. Varying network structures or hyperparameters introduces confounding factors that make it difficult to attribute performance differences to the algorithm itself.
> > > > >
> > > > > Third, research papers should provide reproducible results. Consistent settings across methods make it easier for others to reproduce our findings and confirm our conclusions.
> > > > >
> > > > >
> > > > > All the methods were tested using five random seeds, as done in [1-3].
> > > > >
> > > > > [1] Liu, Yuntao, et al. "Heterogeneous skill learning for multi-agent tasks." Advances in Neural Information Processing Systems 35 (2022): 37011-37023.
> > > > >
> > > > > [2] Liu, Zichuan, Yuanyang Zhu, and Chunlin Chen. "NA
> > > > >  Q: Neural Attention Additive Model for Interpretable Multi-Agent Q-Learning." International Conference on Machine Learning. PMLR, 2023.
> > > > >
> > > > > [3] Na, Hyungho, Yunkyeong Seo, and Il-chul Moon. "Efficient Episodic Memory Utilization of Cooperative Multi-Agent Reinforcement Learning." The Twelfth International Conference on Learning Representations.

---

> ### Comment · Area_Chair_rfki · 2024-11-25
> **Please read rebuttal**
>
> Dear Reviewer 6aVA, Could you please read the authors' rebuttal and give them feedback at your earliest convenience? Thanks. AC

---

### Official Review · Reviewer_17Sh · 2024-11-02

**Soundness:** 3
**Presentation:** 3
**Contribution:** 3
**Rating:** 6
**Confidence:** 4

**Summary:**

TEE promotes agent diversity in multi-agent RL by learning a contrastive trajectory representation and using it to compute a particle-based entropy of the agent trajectories as intrinsic rewards. Combined with QMIX, TEE is evaluated in two multi-agent domains, showing increased diversity and superior empirical performances as a result.

**Strengths:**

- The writing is overall clear and well-organized despite some small issues, which I will mention in other sections. Figures in this paper are helpful as well, although the authors might want to change some font colors in Figure 1 to make it more readable.
- The method is interesting and novel to my knowledge, addressing issues existing in prior methods under the same topic.
- The experiments in the PacMan environment clearly demonstrate the advantages of promoting agent diversity compared with baselines.
- In SMAC, TEE has significant improvement over baselines in the harder scenes. Even in easier scenes where diversity is not required, TEE still has a small advantage, showing that empirically it can balance the diversity maximization and reward exploitation.
- Overall for empirical evaluations, there are a variaty of baselines and ablations, providing sufficient information about the proposed method.

**Weaknesses:**

- I appreciate the theoretical analysis of why mutual information-based methods can have insufficient exploration in the appendix, but it would be more helpful if the authors could include some insights of the analysis into the main text.
- I would like to hear more explanation on why using $Q_a(o_t^a,u_t^a)$ as an input of $Q^{entropy}_a$ can lead to trajectory entropy maximization. This is not straightforward to me. Yes, there will be a gradient passed to $Q_a(o_t^a,u_t^a)$, but why can this gradient make $Q_a$ assign higher values to actions that have higher trajectory entropies?
- TEE is evaluated in two domains, PacMan and SMAC. PacMan to me is more of a demonstrative domain, where behavioral diversity is vital to get the maximum reward. As such, the diversity in evaluated domains feels a bit lacking.
- In the evaluations, TEE is only combined with QMIX. I'm wondering why it's not tested with other MARL algorithms as QMIX is one of the less intuitive one to combine with.

Minor writing issues:
- Using same characters for encoder $g_{\theta_e}$ and $g_{\theta_g}$ is confusing. It feels like two same networks with different parameters.
- L184 the $v$ in $v_a^k$ is italic.
- L248 you state that $Q^{entropy}_{a}$ is "shared". Why does it still have a subscript $a$?

**Questions:**

See other sections.

---

> ### Author Response · Authors · 2024-11-24
>
> Thank you for your detailed review and constructive feedback. Here are the responses to your concerns and questions:
>
> Weakness 1: I appreciate the ... the analysis into the main text.
>
> Thank you for your suggestion. We have made some brief analysis of the limitation of mutual information-based methods that agents are likely to prefer known trajectories, containing more identity information, than novel trajectories, in the first paragraph of Section 3 in our paper.
>
> Weakness 2: I would like to ... higher trajectory entropies?
>
> From Equation 8 in our paper, we note that the intrinsic utility network $Q_a^{entropy}$ is trained by minimizing the TD loss $\mathcal{L}\_{TD}^{entropy}$ toward maximizing the intrinsic rewards $r\_{entropy}^a$. Minimizing the $\mathcal{L}\_{TD}^{entropy}$ also trains the agent utility network $Q_a$ toward maximizing the intrinsic rewards $r\_{entropy}^a$ since the intrinsic utility network $Q_a^{entropy}$ takes a differentiable agent utility $Q_a(o_t^a, u_t^a)$ as the input. Thus, the TD loss $\mathcal{L}\_{TD}^{entropy}$ can be seen as a regularizer for the agent utility network $Q_a$, as shown in Equation 9, which encourages agents to visit trajectories with large entropy.
>
> Weakness 3: TEE is ... a bit lacking.
>
> In addition to Pac-Men and SMAC, we also use SMACv2, which is a totally different benchmark incorporating random start positions and random unit compositions in combat scenarios of StarCraft II. It has a different objective compared to SMAC. SMAC focuses on learning management skills while SMACv2 evaluates the adaptability of MARL methods to changes in environments.
>
> We additionally evaluate our method in Google Research Football (GRF). GRF provides a challenging, physics-based environment that simulates a football game where agents need to learn strategic planning, coordination, and precise timing to succeed. We test our method in three scenarios of GRF. The results are shown in Table 1 in our paper. Compared to baseline methods, our method maintains its outperformance across all scenarios.
>
> Weakness 4: In the evaluations, TEE ... combine with.
>
> Due to space limitations, the details of integrating our method with MAPPO are provided in Appendix D, as referenced in the last paragraph of Section 3.3. We compare TEE+MAPPO against MAPPO across four benchmarks: Pac-Men, SMAC, SMACv2, and GRF. To ensure a fair comparison, consistent network structures and hyperparameters are used. The results, presented in Table 1 in the Appendix, demonstrate that TEE+MAPPO consistently outperforms MAPPO across all tasks.
>
> $g_{\theta_e}$ and $g_{\theta_e}$ are different networks. We both use the character $g$ for them because we need to denote the overall trajectory encoder as $g_{\theta}=\\{g_{\theta_e} ,g_{\theta_g}\\}$.
>
> Yes, $v$ in equation 2 should be italic. Thank you for your careful review.
>
> The parameters of the network $Q_a^{entropy}$ are shared. However, like $Q_a$, we need to learn $Q_a^{entropy}$ for each agent $a$ to maximize the intrinsic rewards. Therefore, $Q_a^{entropy}$ has a subscript $a$.
>
> We hope the responses above have addressed your concerns. We would appreciate receiving your feedback.

---

> > ### Comment · Reviewer_17Sh · 2024-12-02
> >
> > Thank you for your response. I appreciate the explanations and additional experiments. Most of my concerns have been addressed.
> >
> > Following Weakness 2,
> > >Minimizing the $L_{TD}^{entropy}$  also trains the agent utility network $Q_a$ toward maximizing the intrinsic rewards
> > $r_{entropy}^{a}$ since the intrinsic utility network $Q_{a}^{entropy}$ takes a differentiable agent utility $Q_a(o_t^a, u_t^a)$ as the input.
> >
> > This is the exact statement that I don't find straightforward. From my understanding, $Q_a(o_t^a, u_t^a)$ can be $-Q_a^{entropy}$ upon convergence, which is the exact opposite of what you want to achieve.
> >
> > Considering also the evaluation discrepancy brought up by Reviewer 6aVA, I'll maintain my score for now.

---

> > > ### Author Response · Authors · 2024-12-02
> > >
> > > Thank you for the further response. For better clarification, could you further explain why you think $Q_a(o_t^a, u_t^a)$  can be $-Q_a^{entropy}$ upon convergence.

---

> > > ### Author Response · Authors · 2024-12-03
> > >
> > > As the reviewer did not respond to our concerns, we suspect that the reviewer may think the trajectory entropy exploration objective $Q_a^{entropy}$ conflicts with the exploitation objective $Q_a(o_t^a, u_t^a)$.
> > >
> > > From equation 9, our method introduces an auxiliary gradient weighted by $\beta$ for the main objective that maximizes the team rewards, which promotes efficient exploration and would not prevent the agent from learning optimal policies. This is because the parameter $\beta$ ensures that the influence of the entropy-based intrinsic rewards is limited and complementary to QMIX's primary goal. The auxiliary gradient operates as a secondary optimization objective designed to enhance exploration by encouraging diversity among agent trajectories. Moreover, although each agent selects the action with the highest utility $Q_a$, however,  the utility has no actual meaning or constraints [4]. Thus, we can safely integrate an auxiliary gradient to the utility network. This auxiliary gradient aligns with the overall optimization process and does not hinder the maximization of team rewards. Empirical evidence from experiments shows that the integration improves exploration and team performance without disrupting QMIX's ability to optimize for team rewards. In reinforcement learning, efficient exploration has been shown to prevent agents from learning sub-optimal policies. In our method, the trajectory entropy maximization objective encourages agents to thoroughly explore the environment to visit states where they might obtain rewards, resulting in optimal policies.
> > >
> > >
> > > The evaluation discrepancy, as noted in many important works [1-3], stems from different hyperparameters and network structures adopted in different works. In our work, we just need to use consistent hyperparameters and network structures to ensure a fair comparison, as done in [1-3]. The reasons are as follows:
> > >
> > > First, hyperparameters and network structures significantly influence MARL performance. By standardizing them, we ensure that observed performance differences arise from the methods themselves rather than extraneous factors. The key contribution of a MARL paper typically lies in the algorithmic innovation. Keeping all other factors fixed highlights the method's true capabilities without being clouded by secondary influences like architecture design or parameter tuning.
> > >
> > > Second, when testing different algorithms, the goal is to understand their relative effectiveness. Varying network structures or hyperparameters introduces confounding factors that make it difficult to attribute performance differences to the algorithm itself.
> > >
> > > Third, research papers should provide reproducible results. Consistent settings across methods make it easier for others to reproduce our findings and confirm our conclusions.
> > >
> > > [1] Liu, Yuntao, et al. "Heterogeneous skill learning for multi-agent tasks." Advances in Neural Information Processing Systems 35 (2022): 37011-37023.
> > >
> > > [2] Liu, Zichuan, Yuanyang Zhu, and Chunlin Chen. "NA
> > >  Q: Neural Attention Additive Model for Interpretable Multi-Agent Q-Learning." International Conference on Machine Learning. PMLR, 2023.
> > >
> > > [3] Na, Hyungho, Yunkyeong Seo, and Il-chul Moon. "Efficient Episodic Memory Utilization of Cooperative Multi-Agent Reinforcement Learning." The Twelfth International Conference on Learning Representations.
> > >
> > > [4] Rashid, Tabish, et al. "QMIX: Monotonic Value Function Factorisation for Deep Multi-Agent Reinforcement Learning." International Conference on Machine Learning. PMLR, 2018.

---

> ### Comment · Area_Chair_rfki · 2024-11-25
> **Please read rebuttal**
>
> Dear Reviewer 17Sh, Could you please read the authors' rebuttal and give them feedback at your earliest convenience? Thanks. AC

---

> ### Author Response · Authors · 2024-12-01
>
> We hope the responses provided have addressed your concerns and would greatly appreciate your feedback.

---

### Official Review · Reviewer_1kFd · 2024-11-03

**Soundness:** 3
**Presentation:** 3
**Contribution:** 3
**Rating:** 8
**Confidence:** 3

**Summary:**

This paper introduces a novel multi-agent exploration method, Trajectory Entropy Exploration (TEE), to address the issue of homogeneous agent behaviors resulting from parameter sharing. TEE leverages contrastive learning to encode trajectories and identities into distinguishable embeddings, utilizes a nonparametric particle-based entropy estimator to calculate intrinsic rewards, and integrates seamlessly with existing MARL algorithms such as QMIX and MAPPO. Experiments on the Pac-Men, SMAC, and SMACv2 benchmarks demonstrate its superior effectiveness.

**Strengths:**

- The paper is well-motivated and clearly presented. The method effectively combines maximum entropy with contrastive learning to address the diversity challenge in MARL, demonstrating strong empirical performance.
- The tricks used are both practical and easy to implement, and the authors provide thorough ablation experiments to support their approach.

**Weaknesses:**

- Lack challenging MARL tasks that require diversity strategies.

**Questions:**

- As mentioned in the introduction, football games require agents to learn diverse strategies, and Google Research Football is a widely used MARL benchmark. This approach should include additional experiments on this benchmark, similar to the baselines SCDS and FoX.
- I believe it should become a community standard to adopt the evaluation protocol of [Gorsanne et al.](https://arxiv.org/abs/2209.10485) in the multi-agent setting. Not adhering to these standard practices, in my view, weakens what could otherwise be a compelling case for the effectiveness of these algorithms.
- Figure 1 is somewhat unclear. Are the agents' policies derived from the utility functions $Q_a$ or from the combined term $Q_a + \beta Q_a^{entropy}$? If it’s the latter, why not directly update $Q_a$ with $r^a + r^_{entropy}$ instead of using a separate intrinsic utility network?
- Minor items: In Line 184, should $v_a^k$ be written as $\text{v}_a^k$?

---

> ### Author Response · Authors · 2024-11-24
>
> We sincerely appreciate your thorough review and valuable feedback on our paper. We answer your questions below:
>
> Q1: As mentioned ... SCDS and FoX.
>
> Thank you for your suggestion. We additionally evaluate our method in three scenarios of Google Research Football (GRF). The results are shown in Table 1 in our paper. Our method outperforms baseline methods across all scenarios.
>
> Q2: I believe ... of these algorithms.
>
> Using a standardized performance evaluation protocol for cooperative MARL is very important. Our evaluation procedure follows the standardized performance evaluation protocol proposed by Gorsanne et al. We report experimental details in Appendix H and release the code of method in the supplementary material. We set the evaluation interval to 10K steps followed by 32 test episodes. Moreover, we test the generalization of our method in the SMACv2 benchmark, where MARL methods have to adapt to changing scenarios with random unit types and random starting positions. For a fair comparison, we use the same network structures and consistent hyperparameters for all baselines. For generality, we report both the mean and standard deviation of performance results, averaged over five random seeds.
>
>
> Q3: Figure 1 ... utility network?
>
> As illustrated in Figure 1 in our paper, the agents' policies are derived from the combined term $Q_a + \beta Q_{entropy}^a$. We cannot directly update $Q_a$ with $r^a + r_{entropy}^a$. In QMIX, the policies of all agents are co-trained by minimizing the TD loss with a shared team reward (See Appendix C in our paper for more details). Updating $Q_a$ with $r^a + r_{entropy}^a$ may conflict with the main cooperative objective, resulting in unstable results. To support our argument, we design a variant that updates $Q_a$ with $r^a + r_{entropy}^a$ and test it in three super hard scenarios of SMAC. The results are shown below:
>
> |  Method   | 6h\_vs\_8z  |corridor  |3s5z\_vs\_3s6z  |
> |---------------------------------------------|-------------------|------------------|------------------|
> | TEE (intrinsic utility network) | 0.85 $\pm$ 0.03 |0.90 $\pm$ 0.03 |0.87 $\pm$ 0.04 |
> | TEE w/ $r^a + r_{entropy}^a$ | 0.65 $\pm$ 0.08 |0.73 $\pm$ 0.09 |0.49 $\pm$ 0.12|
>
> We note that updating $Q_a$ with $r^a + r_{entropy}^a$ significantly hurt the agent performance. Our method, using the intrinsic utility network $Q_a^{entropy}$ to introduce an auxiliary gradient to the utility network $Q_a$ toward maximizing the intrinsic rewards, achieves more stable results. The intrinsic utility network $Q_a^{entropy}$ allows for each agent's intrinsic reward $r_{entropy}^a$ to be used independently of the shared global environmental reward $r$ in QMIX. This separation ensures that intrinsic motivation (for trajectory entropy maximization) does not interfere with the shared team optimization objective​.
>
> $v_a^k$ should be ltalic. We have corrected the mistake.
>
> We hope to hear from you soon and thank you again for your review.

---

> ### Comment · Area_Chair_rfki · 2024-11-25
> **Please read rebuttal**
>
> Dear Reviewer 1kFD, Could you please read the authors' rebuttal and give them feedback at your earliest convenience? Thanks. AC

---

> > ### Comment · Reviewer_1kFd · 2024-11-27
> >
> > Thank you for the response. The additional experiments are insightful. Based on the revisions made to the paper, I have updated my score.

---

### Official Review · Reviewer_posU · 2024-11-04

**Soundness:** 3
**Presentation:** 2
**Contribution:** 3
**Rating:** 8
**Confidence:** 3

**Summary:**

This paper introduces Trajectory Entropy Exploration (TEE), a multi-agent exploration method that maximizes the entropy of different agent's trajectories in a contrastive trajectory representation space. Prior approaches that incentivize exploration by maximizing the mutual information between the agents' identities and their trajectories can lead to insufficient exploration, where each agent revisits familiar trajectories. In contrast, TEE trains a trajectory representation using *contrastive learning on trajectories and agent identities*, where trajectories from the same agent are pulled closer. This latent representation is then used in a *particle-based entropy estimator*, which estimates the entropy by averaging the distance of each particle to its top $k$ nearest neighbors in the representation space. The proposed method is instantiated with the QMIX algorithm, where an additional $Q$-value for the entropy rewards is learned and added to the standard $Q$-value. TEE exhibits strong empirical results across a suite of MARL environments including Pac-Men, SMAC, and SMAC 2, outperforming baselines in exploration and sample efficiency. The design choices in TEE are thoroughly validated via exhaustive ablation studies.

**Strengths:**

1. The paper is motivated by carefully analyzing existing multi-agent exploration methods and identifying a potential degeneracy in those methods, i.e., that maximizing the mutual information between agent identities and trajectories can lead to insufficient exploration. The proposed method -- maximizing the entropy of the mixture of trajectories while being informative about agent identities -- is a convincing solution.
2. TEE demonstrates strong empirical results on a suite of challenging multi-agent RL environments, significantly outperforming standard MARL and MI-based exploration methods. Figure 2 in particular shows that TEE fully explores all four rooms in the Pac-Men environment whereas Q-MIX only explores two of the four rooms. Figure 5 further shows that TEE explores the map in SMAC more thoroughly than MI-based exploration methods. These lend support to the effectiveness of the exploration method.
3. The design choices in TEE are backed by thorough ablation studies in Section 4.3, Figure 6, and Table 1. The authors show the importance of parameterizing the trajectory encoder using an autoregressive model, learning the agent identities instead of fixing to one-hot, and so on.

**Weaknesses:**

1. The proposed method relies on a nonparametric entropy estimation that introduces additional hyperparameters, e.g. the number of neighbors $k$. This procedure also introduces additional computational overhead, since it needs to run batch KNN in each training step.
2. Some design choices of the method are not straightforward. See questions 1, 2, and 3 below.
2. The presentation of the paper can be improved. Some paragraphs are too dense (e.g. Section 4.3) and figures need more informative captions.

**Questions:**

1. If the limitation of methods maximizing the mutual information between trajectories and agent identities is insufficient exploration, why can't we add a max entropy term to each agent's objective, so that they explore as much as possible while being distinguishable from each other? Are there related works that do this?
2. The autoregressive trajectory encoder is trained on trajectories but only applied to a single state when estimating the particle-based entropy (according to Algorithm 1). So in principle one only needs a state encoder. Why do you train a trajectory encoder when it's only applied to states?
3. What happens if you replace the L2 distance in the entropy estimator with negative cosine similarity? Since the encoder is trained with contrastive loss, cosine sim seems a more natural metric in the latent space.
4. In scenarios requiring homogeneous behavior (Appendix. F), do you need to tune the coefficient $\beta$ of the entropy loss?
5. Section 3.1 on learning contrastive trajectory representations is disconnected from Sections 3.2 and 3.3 on the particle-based entropy estimator and its integration with MARL algorithms. I suggest adding a paragraph in Sec. 3.3 stating that the representation is learned jointly with the policy, to provide a more complete picture.
6. I would also suggest adding a more detailed description of TEE in the caption of Figure 1 to make it self-contained.

---

> ### Author Response · Authors · 2024-11-24
>
> Thank you for your careful review and for providing us with detailed and helpful feedback. We respond to your concerns below:
>
> Weakness 1: The ...  step.
>
> The performance of our method is not sensitive to the value of $k$, as demonstrated in Appendix I. Finding the $k$ nearest neighbors (KNN) does not introduce high computational overhead in our method. This is because, in multi-agent settings, we only have a total number of $|A|$ data points, where $|A|$ is the number of agents. In most benchmarks, $|A|$ is quite small.
>
> Q1: If the limitation ... this?
>
> Adding a max entropy term to each agent's objective to encourage policies to choose actions with higher uncertainty cannot efficiently solve the limitation of the mutual information-based methods. First, the agent still overfits to the known trajectories due to the existence of the mutual information objective, impeding further exploration. Second, while maximizing policy entropy has been applied successfully in single-agent RL, such as SAC [1], it may not perform effectively in multi-agent settings. To prove our argument, we design a variant of QMIX that incorporates a mutual information-based trajectory discriminator $p(i|\tau)$ and an entropy term $H(\pi(\cdot|s_t)) = -\sum_{a} \pi(a|s_t) \log(\pi(a|s_t))$. To calculate the entropy term, we use a softmax operator after the output layer of the utility network to output a probability distribution over actions. We evaluate this variant in three super hard scenarios of SMAC. The evaluation results are shown below:
>
> |  Method   | 6h\_vs\_8z  |corridor  |3s5z\_vs\_3s6z  |
> |---------------------------------------------|-------------------|------------------|------------------|
> | TEE (trajectory entropy maximization) | 0.85 $\pm$ 0.03 |0.90 $\pm$ 0.03 |0.87 $\pm$ 0.04 |
> | TEE w/ $p(i\|\tau)$ and $H(\pi(\cdot\|s_t)) = -\sum_{a} \pi(a\|s_t) \log(\pi(a\|s_t))$ | 0.49 $\pm$ 0.05 |0.65 $\pm$ 0.09 |0.53 $\pm$ 0.06|
>
> The results show that our method significantly outperforms the variant, demonstrating the effectiveness of our trajectory entropy maximization objective. We suspect this is because the agents still overfit to known trajectories although the entropy term increases the uncertainty in action selections.
>
> Q2: The autoregressive ... states?
>
> We use the trajectory encoder $g_{\theta}=\\{g_{\theta_e} ,g_{\theta_g}\\}$ to encode the whole trajectories when estimating the particle-based entropy. We encode the whole trajectory literately from $t=0$ to $t=T$. $T$ is the length of the trajectory. We do not need to include the output from the autoregressive model at the previous time step as the input to the encoder $g_{\theta}$. Thus, the encoder $g_{\theta}$ solely uses the current observation $o_t$ as the input. Actually, the autoregressive model $g_{\theta_g}$ continues to incorporate prior observations.
>
>
> Q3: What happens ... space.
>
> To evaluate the effectiveness of the negative cosine similarity, we design a variant that ablates the L2 distance with the negative cosine similarity. We test this variant in three super hard scenarios of SMAC. The results are shown below:
>
> |  Method   | 6h\_vs\_8z  |corridor  |3s5z\_vs\_3s6z  |
> |---------------------------------------------|-------------------|------------------|------------------|
> | TEE (L2 distance) | 0.85 $\pm$ 0.03 |0.90 $\pm$ 0.03 |0.87 $\pm$ 0.04 |
> | TEE w/ negative cosine similarity | 0.67 $\pm$ 0.03 |0.71 $\pm$ 0.06 |0.58 $\pm$ 0.08|
>
> Our method, using L2 distance, maintains its outperformance across the three scenarios. Cosine similarity is often used in scenarios where vector direction or relative orientation matters more than magnitude. The L2 distance is more efficient in measuring the distance between different trajectories in the latent space.
>
>
> Q4: In scenarios ... entropy loss?
>
> No, we do not need to specifically tune the coefficient $\beta$. Our method can efficiently balance exploration and exploitation. The values of $\beta$ adopted in our method are provided in Table 2.
>
> Q5:Section 3.1 ... picture.
>
> We have added a paragraph in Section 3.3 to connect it with Section 3.1: During training, we alternately train the trajectory encoder and policies of agents. We first sample trajectories from the replay buffer to train the encoder to learn distinguishable trajectory representations by minimizing the contrastive learning loss. Then we calculate the particle-based trajectory entropy estimator based on the learned trajectory representations for policy learning.
>
> Q6: I would ... self-contained.
>
> We have updated our paper and added a more detailed description of our method in the caption of Figure 1.
>
> We hope all your problems have been solved, and thank you again for taking the time to review our paper.
>
> [1] Haarnoja, Tuomas, et al. "Soft actor-critic algorithms and applications." arXiv preprint arXiv:1812.05905 (2018).
>
> [2] Mahajan, Anuj, et al. "Maven: Multi-agent variational exploration." Advances in neural information processing systems 32 (2019).

---

> > ### Comment · Reviewer_posU · 2024-11-25
> >
> > Thanks for addressing my comments. I appreciate the additional baselines and ablations, as well as the clarification on the autoregressive trajectory encoder. I will increase my score.

---

### Meta-Review · Area_Chair_rfki · 2024-12-17

**Metareview:**

This paper introduces a trajectory entropy augmented multi-agent exploration strategy. The proposed approach is based on QMIX and achieved in the trajectory latent space with a contrastive objective. While the idea itself is innovative, the manuscript does not include enough domains or perform sufficient ablative studies. The test domains are relatively simple. Moreover, some reviewers suggest adding the proposed approach to policy-based backbones, which would demonstrate the generality of TEE. The overall clarity needs improvement, but it is minor. In summary, this paper is above the bar of ICLR. We encourage the authors to further improve the quality for the camera-ready version.

**Additional Comments On Reviewer Discussion:**

The authors addressed some of the concerns of the reviewers.

---

### Decision · Program_Chairs · 2025-01-22

Accept (Poster)